

**Nitrogen input [15]N-signatures are reflected in plant [15]N natural abundances of sub-tropical**
**forests in China**
Geshere Abdisa Gurmesa [a,b,c], Xiankai Lu[a], Per Gundersen[b], Yunting Fang[d], Qinggong Mao[a], Chen
Hao[e], Jiangming Mo[a]
*[a] Key Laboratory of Vegetation Restoration and Management of Degraded Ecosystems and Guangdong Provincial Key*
*Laboratory of Applied Botany, South China Botanical Garden, Chinese Academy of Sciences, Guangzhou 510650,*
*China*
*[b] Department of Geosciences and Natural Resource Management, University of Copenhagen, Copenhagen, Denmark*
*[c] Sino-Danish Center for Education and Research, Aarhus, Denmark*
*[d] State Key Laboratory of Forest and Soil Ecology, Institute of Applied Ecology, Chinese Academy of Sciences,*
*Shenyang 110164, China*
*[e] Huanjiang Observation and Research Station for Karst Ecosystem, Key Laboratory of Agro-ecological Processes in*
*Subtropical Region, Institute of Subtropical Agriculture, Chinese Academy of Sciences, Changsha 410125, China*
Corresponding author:
Jiangming Mo
E-mail: mojm@scib.ac.cn



**Abstract**
Natural abundance of $^{15}$N ($\delta^{15}$N) in plants and soils can provide integrated information on
ecosystem nitrogen (N) cycling, but it has not been well tested in warm and humid sub-tropical
forests. In this study, we examined the measurement of $\delta^{15}$N for its ability to assess changes in N
cycling due to increased N deposition in an old-growth broadleaved forest and a secondary pine
forest in a high N deposition area in southern China. We measured $\delta^{15}$N of inorganic N in input and
output fluxes under ambient N deposition, and N concentration (N%) and $\delta^{15}$N of major ecosystem
compartments under ambient and after decadal N addition at 50 kg N ha$^{-1}$yr$^{-1}$. Our results showed
that the N deposition was $\delta^{15}$N-depleted (-12 ‰) mainly due to high input of depleted $NH_4^+$-N.
Plant leafs in both forest were also $\delta^{15}$N-depleted (-4 to -6‰). The old-growth forest had higher
plant and soil N%, and was more $^{15}$N-enriched in most ecosystem compartments relative to the pine
forest. Nitrogen addition did not significantly affect N% in both forests, indicating that the
ecosystem pools are already N-rich. Soil $\delta^{15}$N was not changed significantly by the N addition in
both forests. However, the N addition significantly increased the $\delta^{15}$N of plants toward the $^{15}$N
signature of the added N (~0 ‰), indicating incorporation of added N into plants. Thus, plant $\delta^{15}$N
was sensitive to ecosystem N input manipulation although N% was unchanged in these N-rich sub-
tropical forests. We interpret the depleted $\delta^{15}$N values of plants as an imprint from the high and
$\delta^{15}$N-depleted N deposition. The signal from the input (deposition or N addition) may override the
enrichment effects of fractionation during the steps of N cycling that are observed in most warm
and humid forests. Thus, interpretation of ecosystem $\delta^{15}$N values from high N deposition regions
need to include data on the deposition $\delta^{15}$N signal.

Key words: Natural $^{15}$N abundance, N addition, N deposition, sub-tropical, China



## 1. Introduction

Nitrogen (N) deposition onto terrestrial ecosystems has dramatically increased due to anthropogenic activities (Galloway, 2005), and recently, the increase has been strong in the warm and humid parts of south Eastern Asia, particularly in China (Fang et al., 2011a). Nitrogen deposition onto forests that exceeds plant and microbial demand may increase nutrient leaching and soil acidification (Lu et al., 2014), and potentially causes nutritional imbalances in the forests (Schulze, 1989). Studies of fates and process responses to increased N deposition using coordinated N addition experiments in temperate and boreal forests show that the effects of increased N deposition largely depend on the initial N status of the forests (Gundersen et al., 1998; Hyvönen et al., 2008). Accordingly, N limited forests that often show a growth response to added N, largely retain the deposited N, whereas N saturated forests, with N input in excess of plant and microbial demand, lose N through leaching and denitrification. Although some studies from (sub) tropical regions also suggest that N leaching from tropical forests is related to the initial N status of the forests (Chen and Mulder, 2007; Fang et al., 2009), the observations are far from being conclusive, especially in regions that are subjected to increased anthropogenic N deposition (Townsend et al., 2011).

Natural abundance of $^{15}$N (the ratio of $^{15}$N:$^{14}$N, and given as per mil; $\delta^{15}$N) of ecosystem has been suggested to provide integrated information about effects of both current and past N cycling in ecosystems (Handley and Raven 1992; Robinson, 2001). Using the $\delta^{15}$N data to interpret N status and changes in N cycling of an ecosystem rely on the fractionation that occurs during N transformation processes and creates differences in $\delta^{15}$N between substrate and product pools (Högberg, 1997). Accordingly, it was suggested that N-saturated ecosystems are characterized by elevated $\delta^{15}$N above the atmospheric standard (i.e., 0 ‰) due to increased rate of N cycling (e.g., mineralization and nitrification), and losses of $^{15}$N-depleted N-forms by leaching and denitrification. Global data indeed show elevated leaf $\delta^{15}$N at N rich conditions, i.e. increasing leaf





$\delta^{15}N$ with increasing leaf N concentration and higher leaf $\delta^{15}N$ in warmer climates (Craine et al.
2009). Hence, Martinelli et al. (1999) also found tropical forests to have higher leaf $\delta^{15}N$ than
temperate forests. However, the influence of increased N deposition on the $\delta^{15}N$ levels is not well
known. For example, even though plant $\delta^{15}N$ could increase with N deposition (Emmett et al.,
1998), it may not be the case across all regions where not only ecosystem N status but also a region-
specific $^{15}N$ signature of deposited N may influence ecosystem $\delta^{15}N$ (Fang et al., 2011b; Pardo et
al., 2006). Moreover, interpretation of ecosystem $\delta^{15}N$ is hampered by the uncertainties in $\delta^{15}N$ of
plant N sources, the magnitude of isotopic fractionations during N transformation processes, and the
complex behavior of $^{15}N$ in soils and plants (Robinson 2001). On the other hand, our understanding
of ecosystem $\delta^{15}N$ is primarily based on studies from North America, Europe, South America, and
Australia (Amundson et al., 2003; Craine et al., 2009; Martinelli et al., 1999). Apparently, there is a
data gap over large areas in eastern Asia, particularly sub-tropical China, which is among the hot-
spots for N deposition.
Plant leaf and soil $\delta^{15}N$ values are most commonly used to assess N status and changes in N
cycling rates, and other ecosystem pools are neglected or rarely measured. The turn-over times of N
pools vary among different ecosystem compartments, and thus their $\delta^{15}N$ may respond differently to
specific disturbances. For example, within plant compartments, small active N pools such as leaves
reflect recent N cycling whereas the larger N pools such as wood might reflect long term-changes in
N cycling. Nevertheless, studies that measured $\delta^{15}N$ values of all major ecosystem pools are rare
(e.g. Liu, 1995), emphasizing the need for more rigorous studies to provide complete patterns of
$\delta^{15}N$ in the leaf-to-soil continuum, and their response to N input manipulation, especially in the
tropical forests.
We evaluated $\delta^{15}N$ of tropical forests, and its response to increased N deposition using long-
term N addition experimental plots established in 2003 in an old-growth broadleaf forest and a pine



plantation forest in the Dinghushan Biosphere Reserve (DHSBR) in southern China (Mo et al.,
2006). The old-growth forest is more N-rich, and has less N retention capacity than the pine forest
(Fang et al., 2006). Nitrogen addition studies in these forests documented that increased N input
causes increased N leaching (Fang et al., 2008, 2009), $N_2O$ emission (Zhang et al., 2008) and soil
acidification (Lu et al., 2014). Here, our objectives are (1) to determine $\delta^{15}N$ of ecosystem
compartments across the leaf-to-soil continuum in the two forest types, (2) to assess response of
$\delta^{15}N$ in major ecosystem pools to decadal N addition in the two forests. We hypothesized that i)
$\delta^{15}N$ of plants and soil in these forests will be higher (more $^{15}N$-enriched) than those reported in
temperate forests due to high N status at DHSBR, and subsequent fractionation during leaching and
denitrification, ii) decadal N addition would change plant and soil $\delta^{15}N$ towards the $^{15}N$ signature of
the added N due to its incorporation into ecosystem pools, and the changes would differ between
the two forests due to their difference N status and N cycling rates.

**2. Methods**
**2.1. Study site**
The study was conducted in the Dinghushan Biosphere Reserve, southern China (112$^{o}$33' E and
23$^{o}$10' N) with typical sub-tropical monsoon climate. Mean annual temperature and mean annual
precipitation were 22.2 C$^{o}$ and 1927 mm, respectively. The reserve DHSBR has been experiencing
high rates of atmospheric N deposition (21-38 kg N ha$^{-1}$ yr$^{-1}$ as inorganic N in bulk precipitation)
since 1990's (Fang et al., 2008). In 2009 to 2010, total wet N deposition was 34.4 kg N ha$^{-1}$ yr$^{-1}$ (Lu
et al., 2013). We used two common forest types that grow on the relatively steep slopes in the
reserve; an old-growth broad-leaved forest (hereafter named as BF) and a pine plantation forest
(hereafter named as PF) (Mo et al., 2006). The BF is a regional climax mixed broad-leaved forest,
which has been protected for at least the last 400 years with minimum human disturbances (Shen et
al., 1999). The PF was planted after clear-cut of the original climax forest in the 1930s and has been
subjected to human disturbances such as litter and shrub harvesting until the recent past (Mo et al.,

2005).


Table 1. Selected characteristics of the mineral soil (0-10 cm) in the two forest types. Data on soil bulk
density, total P and extractable $NH_4^+$-N are obtained from Fang et al., (2006). Values given in parenthesis
indicate SE ($n = 3$).

| Parameters | Broad-leaved forest (BF) | Pine forest (PF) |
|---|---|---|
| Bulk density (g cm$^{-3}$) | 0.9 (0.03) | 1.3 (0.03) |
| pH (H$_2$O) | 3.8 (0.02) | 4.0 (0.04) |
| C concentration (%) | 3.8 (0.80) | 1.8 (0.03) |
| N concentration (%) | 0.3 (0.04) | 0.1 (0.01) |
| C/N ratio | 13.6 (0.9) | 13.9 (0.7) |
| Total P (mg kg$^{-1}$) | 59 (3) | 43 (3) |
| Extractable $NH_4^+$-N (mg kg$^{-1}$) | 2.1 | 3.3 |
| Extractable $NO_3^-$-N (mg kg$^{-1}$) | 12.7 | 2.6 |


The major canopy species in the BF were *Castanopsis chinensis*, *Machilus chinensis,*
*Schima superba*, *Cryptocarya chinensis*, and *Syzygium rehderianum* and the most common
understory species is *Hemigramma decurrins. Pinus massoniana* and *Dicranopteris dichotoma* are
the dominant tree and understory species in the PF, respectively. The soil in the reserve is classified
as Lateritic Red Earth (Oxisol) formed from Devonian sandstone and shale with a thin layer of
forest floor litter (0.5-3.0 cm), but the soil depth is variable ranging from 30 cm in the PF to more
than 60 cm in the BF. Probably due to erosion after the clear-cut and the continued human
disturbance the PF had lower total soil carbon, N and phosphorus (P) content than the BF (Table 1).





### 2.2. Experimental design

We used an ongoing N addition experiment established in both forests in July 2003 (Mo et al., 2006). The experimental plots consist of control plots and N addition treatments at 50 kg N ha$^{-1}$ yr$^{-1}$ and 100 kg N ha$^{-1}$ yr$^{-1}$ each with three replicates in both forests, and an additional 150 kg N ha$^{-1}$ yr$^{-1}$ in the BF. However, in this study we used only the control plots and the low-N treatment plots at 50 kg N ha$^{-1}$ yr$^{-1}$ (hereafter named as N-plots). Each plot is 10 m x 20 m with at least a 10-m wide buffer strip to the next plot. In the N-plots, $NH_4NO_3$ is mixed with 20 L of water, and is added monthly below the canopy using a backpack sprayer, whereas the control plots received equivalent 20 L water with no fertilizer. The added N has $\delta^{15}N$ of about -3 ‰ on $NH_4^+$-N and about 1.8 ‰ on $NO_3^-$-N, with $\delta^{15}N$ of $NH_4NO_3$ being -0.7 ‰.

### 2.3. Sampling and analysis of plant and soil pools

Major ecosystem compartments, including leaves, twigs, branches, bark and wood of canopy trees, leaves of understory vegetation, fine roots, and 0-30 cm mineral soil were sampled in January 2013 to determine N concentration (%) and the $\delta^{15}N$ (‰) of the forests. A branch per dominant tree species per plot was cut from the height reached using a pole pruner (c. 7-8 m) taking advantage of the steep slope, and was separated into leaves, twigs and small branches. Bark samples were cutoff the dominate trees at breast height using a knife. After removing the bark, wood cores were sampled using an increment borer and separated visually into sapwood (usually the outer 2-3 cm recent wood) and older wood (heartwood). Dominant understory plant species were cut with a knife and kept separate for each species. A total of seven tree species in the canopy/sub-canopy layer and five plant species in the understory layer (young trees, shrubs, herbs and liana) of the BF were sampled, whereas in the PF the dominant pine tree and five species in the understory layer were sampled. Mineral soil samples were taken using an auger (5.1 cm in diameter) and were divided into three



layers (0-10, 10-20, 20-30 cm). Two soil cores were sampled and pooled together to form one
composite sample for each depth per plot. Living fine roots were hand-sorted from the soil samples
for each depth, but combined to one composite sample for the whole profile (0-30 cm) because the
amount of fine roots in each depth was too small to grind and analyze separately. Litterfall was
collected monthly during July-September 2012 and pooled together to make one composite sample
per plot. The litter was sorted in the laboratory into leaf and others (branches, fruits, flowers, barks),
but only leaf values are reported.

All plant and soil samples were oven-dried at 70$^o$C, and ground to a fine and homogeneous

powder. Mineral soils were sieved (2 mm mesh) to remove non-soil materials, air-dried at room
temperatures and milled to fine powder. Subsamples were dried at 105°C, and all results are
reported on 105°C basis. Based on their approximate N%, about 4-5 mg of the samples were
weighed into tin capsules, and $\delta^{15}$N and N concentration of the samples were determined
simultaneously on an isotope ratio mass spectrometer (Isoprime 100, Isoprime Ltd.) coupled to an
automatic, online elemental analyzer (vario ISOTOPE cube). An internal standard needle sample
from temperate forests, which has been analysed in multiple runs at several laboratories, was used
to check reproducibility of the $\delta^{15}$N determination. We analyzed N % and $\delta^{15}$N separately for each
dominant tree species per plot, but compartment mean values are reported. Natural abundance $\delta^{15}$N
in samples was reported in per mil (‰) relative to atmospheric $N_2$.

**2.4.Sampling and analysis of water samples**
Precipitation, throughfall, surface runoff and soil solution (seepage) from 0-20 cm were sampled
monthly from September 2012 to February 2013 (including a dry period in December and January)
in the control plots to assess the $\delta^{15}$N of N input and output in the two forests under ambient N
deposition. Bulk precipitation was collected at an open area close to the experimental site using an



open glass funnel (12 cm in diameter), connected to a 5 L sampling bottle with polypropylene tubes.
Throughfall was collected by PVC pipes at five random points within each plot (intercept area 0.8
$m^2$ for each collector) at about 1.3 m above the ground in each forest. Each collector was connected
to two 50 L buckets with polypropylene tubes. Soil solution from 20 cm depth (seepage water) was
obtained using two zero tension tray lysimeters (755 $cm^2$ per tray) installed in each plot. Each
lysimeter was connected to a 20 L bottle using the steep slope of the sites to facilitate sampling. In
addition, since the plots are situated on steep slopes, one plot in both the pine and broad-leaved
forests was delimited hydrologically by plastic and concrete barriers to sample and quantify surface
runoff.
Natural $^{15}N$ abundance of both $NH_4^+$-N and $NO_3^-$-N in water samples were analyzed after
chemical conversion to nitrous oxide ($N_2O$). The $NH_4^+$-N was initially oxidized to nitrite ($NO_2^-$) by
hypobromite ($BrO^-$) and the $NO_2^-$ is then quantitatively converted into $N_2O$ by hydroxylamine
($NH_2OH$) under strongly acidic conditions (Liu et al., 2014). Similarly, a series of chemical
reactions of vanadium (III) chloride ($VCl_3$) and sodium azide under acidic conditions was used to
convert $NO_3^-$-N into $N_2O$ (Lachouani et al., 2010). The produced $N_2O$ was subsequently analysed
for $^{15}N$ abundance by a purge-and-trap coupled with an isotope ratio mass spectrometer (PT-IRMS)
(Liu et al., 2014).

**2.5. Calculations and Statistics**
To evaluate effects of decadal N addition on the whole ecosystem (plant and soil) N% and $\delta^{15}N$, we
determined N pool weighted plot means of N% and $\delta^{15}N$ using N pools for each compartment
quantified in Gurmesa et al. (2016). We excluded the heartwood and sapwood pools in the plant
pool calculations for two reasons; first the low N content in wood samples caused larger
uncertainties on the $\delta^{15}N$ determinations, and secondly heartwood and a major part of the sapwood



was formed prior to the initiation of the N addition treatment. We expect the later to be the
explanation that particular heartwoods showed opposite effects of N addition compared to all other
compartments.

212   Differences between the two forests in plot mean N% and $\delta^{15}$N of the different ecosystem

compartments in control plots were analysed using $t$-test. The effect of treatment on N% and $\delta^{15}$N
of each tree compartments and understory leaf in each forest was analyzed using mixed model
ANOVA with treatment as explanatory factor and plant species as a random factor, since plant
species differed significantly in both parameters (Gurmesa, 2016). Treatment effect on N % and
$\delta^{15}$N of soil in each layer, litter and fine roots in each forest was analyzed using $t$-test. Effects of
treatment on N pool weighted plot means in each forest, and their differences between the control
plots of the two forests were also analyzed using $t$-test.

221   **3. Results**

222   **3.1.Concentration and $\delta^{15}$N of $NH_4^+$-N and $NO_3^-$-N**

Dissolved $NH_4^+$-N in waters samples, including both inputs through precipitation and throughfall
and outputs fluxes by surface runoff and seepage at 20 cm depth were $^{15}$N-depleted (negative $\delta^{15}$N
values) in both forest (Table 2). Nitrate-N was $^{15}$N-enriched and had positive $\delta^{15}$N in the input
fluxes, but was slightly $^{15}$N-depleted in output fluxes in both forests. Ammonium-N was the
dominant N form in precipitation and throughfall, but $NO_3^-$-N appeared to be the dominant N form
in the output fluxes through surface runoff and leaching below 20 cm soil (**Fig**. S1). Mean $\delta^{15}$N of
both $NH_4^+$-N and $NO_3^-$-N in input and output fluxes did not significantly differ between the two
forests. The $\delta^{15}$N of $NH_4^+$-N in surface runoff and seepage were significantly and positively related
to $\delta^{15}$N of $NH_4^+$-N in throughfall for combined data in the two forests (Fig 1a, b), but the correlation
was not significant for $NO_3^-$-N (Fig 1c, d).





Table 2. Mean $\delta^{15}$N (‰) of $NH_4^+$-N and $NO_3^-$-N in bulk precipitation, throughfall, surface runoff and seepage
at 20 cm depth in control plots from September 2012 to February 2013. Numbers in parenthesis indicate
standard error of the mean (SE) ($n = 3$).

| Fluxes | Broad-leaved forest (BF) | | Pine forest (PF) | |
|---|---|---|---|---|
| | $NH_4^+$-N | $NO_3^-$-N | $NH_4^+$-N | $NO_3^-$-N |
| Precipitation* | -16.6 | 4.1 | -16.6 | 4.1 |
| Throughfall | -15.2 (2.3) | 3.6 (0.2) | -15.5 (1.8) | 2.8 (0.3) |
| Surface runoff | -13.1 (1.7) | -1.9 (0.6) | -9.7 (1.0) | -1.5 (0.6) |
| Seepage | -22.6 (0.9) | -0.9 (1.3) | -21.3 (2.3) | -0.9 (0.2) |

*Precipitation was collected at open area within the reserve, and was assumed to be the same for both forests

### 238  3.2. Ecosystem compartment N% and $\delta^{15}$N

In agreement with our expectation based on results from earlier studies in the DHSBR, the BF is
more N-rich than PF. Nitrogen concentrations of plant compartments were significantly higher in
the BF than in the PF, except in leaves of canopy trees (Table 3). Nitrogen concentration in litter-
fall and fine roots in the BF only marginally differed from that of PF. Soil N% was significantly
higher in the BF at all depths (Table 3).

Natural $\delta^{15}$N of all plant compartments differ significantly between the two forests with the

PF being more $^{15}$N-depleted than the BF (Table 4). Soil $\delta^{15}$N did not show significant difference
between the two forests at any depth (test data not shown), though soil still tended to be more $^{15}$N-
enriched in the BF (Fig. 2).





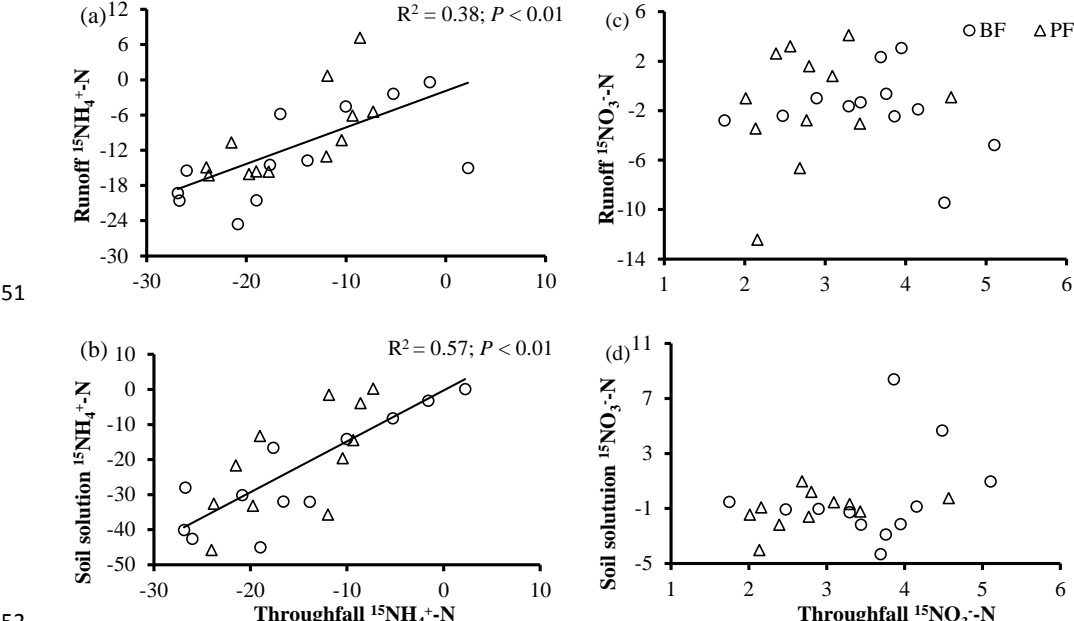

**Fig. 1** Correlation between $\delta^{15}N$ (‰) of $NH_4^+$-N in throughfall and that of $NH_4^+$-N in surface runoff (a), and soil solution (b), and correlation between $\delta^{15}N$ of $NO_3^-$-N in throughfall and that of $NO_3^-$-N in surface runoff (c), and soil solution (d). No significant effect of forest type was detected; thus the regression line shown was based on data from both forests.

### 3.3. Effects of N addition on compartment N% and $\delta^{15}N$

Nitrogen concentrations in all measured plant and soil compartments were not significantly affected by N addition in the BF, except in the sapwood (Table 3). In the PF, N% increased in all ecosystem pools, but the change was significant only in leaf of understory plants (Table 3). Nitrogen concentration of N pool weighted average plant pools calculated per plot did not significantly increased by N addition in both forests, but it slightly increased in the PF (Table 3).

Plant $\delta^{15}N$ was negative in both control and N-plots in both forests, but N addition significantly increased the $\delta^{15}N$ of most plant compartments (Table 4). The changes were more pronounced in the small active plant pools such as leaves of trees and understory plants.



Table 3. Mean N% of different ecosystem pools in the broad-leaved (BF) and pine forests (PF). Values in
parenthesis indicate SE of plot means ($n = 3$). Within each forest type the effect of N addition is shown by $p$-
values from a mixed model with species as random factor. The last column shows $p$-values for a difference
between the ambient plots of the two forests using $t$-test. Bolded $p$-values indicate significant difference.

| Compartment | Broad-leaved forest (BF) | | | Pine forest (PF) | | | Forest type effect |
|---|---|---|---|---|---|---|---|
| | Control | N addition | $p$-values | Control | N addition | $p$-values | $p$-values |
| ***Plants*** | | | | | | | |
| Tree leaf | 1.71 (0.19) | 1.69 (0.19) | 0.48 | 1.44 (0.11) | 1.68 (0.02) | 0.10 | 0.08 |
| Twig | 1.28 (0.19) | 1.17 (0.05) | 0.59 | 0.99 (0.77) | 0.97 (0.08) | 0.79 | **<0.01** |
| Branch | 0.86 (0.15) | 0.81 (0.16) | 0.13 | 0.58 (0.05) | 0.60 (0.06) | 0.87 | **<0.01** |
| Bark | 0.71 (0.16) | 0.7 (0.16) | 0.55 | 0.57 (0.02) | 0.61 (0.05) | 0.52 | **<0.01** |
| Sapwood | 0.27 (0.07) | 0.3 (0.07) | **<0.01** | 0.18 (0.02) | 0.11 (0.02) | 0.07 | **0.02** |
| Heartwood | 0.16 (0.04) | 0.16 (0.03) | 0.28 | 0.06 (0.0) | 0.09 (0.03) | 0.30 | **<0.01** |
| Understory[a] | 2.04 (0.02) | 1.98 (0.17) | 0.09 | 1.61 (0.41) | 1.77 (0.40) | **<0.01** | **<0.01** |
| Fine root | 1.4 (0.16) | 1.81 (0.17) | 0.14 | 0.87 (0.13) | 0.96 (0.04) | 0.43 | 0.06 |
| Litter-fall | 1.56 (0.06) | 1.48 (0.08) | 0.43 | 1. 39 (0.03) | 1.72 (0.11) | 0.56 | 0.09 |
| ***Soil*** | | | | | | | |
| 0-10 cm | 0.27 (0.04) | 0.28 (0.01) | 0.83 | 0.13 (0.01) | 0.12 (0.01) | 0.37 | **0.03** |
| 10-20 cm | 0.18 (0.01) | 0.19 (0.01) | 0.59 | 0.07 (0.00) | 0.06 (0.00) | 0.37 | **<0.01** |
| 20-30 cm | 0.12 (00) | 0.14 (00) | 0.10 | 0.06 (0.00) | 0.05 (0.00) | 0.12 | **<0.01** |

[a] the values are only for leaf of understory plants

However, inconsistent effect of N addition on $\delta^{15}N$ was observed in the wood parts (Table
4). For heartwoods, the effects were significant, but in different directions than other plant pools for
both forests. Due to low N% and challenges in grinding of wood samples it was difficult to get
reliable the $\delta^{15}N$. Also much of the sampled wood was formed prior to the treatment and thus, no





further evaluation was done for the wood samples. Nitrogen addition did not cause significant effect
on $\delta^{15}N$ of litter-fall and fine roots. In the BF, there was no correlation between leaf N% and $\delta^{15}N$,
but a positive correlation was found for the PF as both N% and $\delta^{15}N$ tended to increase in parallel
due to N addition (data not shown).

Table 4. Mean $\delta^{15}N$ (‰) of plant pools in the broad-leaved (BF) and pine forests (PF). Values in parenthesis
indicate SE of plot means ($n = 3$). Within each forest type the effect of N addition is shown by $p$-values from
a mixed model with species as random factor. The last column shows $p$-values for a difference between the
ambient plots of the two forests using $t$-test. Bolded $p$-values indicate significant difference.

| Sample type | Broadleaved forest (BF) | | | Pine forest (PF) | | | Forest type effect |
|---|---|---|---|---|---|---|---|
| | Control | N addition | $p$-values | Control | N addition | $p$-values | $p$-values |
| Tree leaf | -4.0 (0.5) | -3.4 (0.6) | **<0.01** | -5.4 (0.1) | -3.5 (0.3) | **<0.01** | **<0.01** |
| Twigs | -4.3 (0.8) | -3.8 (0.9) | **0.04** | -5.7 (0.1) | -4.0 (0.3) | **<0.01** | **<0.01** |
| Branches | -4.6 (0.4) | -4.1 (0.3) | **<0.01** | -5.7 (0.2) | -4.1(0.6) | 0.08 | **0.03** |
| Barks | -2.8 (0.8) | -2.4 (0.6) | 0.16 | -4.0 (0.4) | -2.6 (0.2) | **0.03** | 0.03 |
| Sapwood | -1.9 (0.5) | -1.8 (0.3) | 0.51 | -0.9 (0.4) | 1.8 (1.6) | 0.18 | 0.06 |
| Heartwood | **-1.6 (0.9)** | **-2.3 (0.9)** | **<0.01** | **3.2 (0.8)** | **-0.71 (1)** | **0.04** | **< 0.01** |
| Understory[a] | -3.6 (0.9) | -2.2 (1.1) | **<0.01** | -5.6 (0.5) | -3.54 (0) | **<0.01** | **<0.01** |
| Fine root | -2.8 (0.6) | -1.7 (0.8) | 0.33 | -5.1 (0.5) | -3.6 (0.3) | 0.06 | **0.04** |
| Litter-fall | -3.9 (0.1) | -3.9 (0.1) | 0.98 | -4.8 (0.2) | -4.0 (0.3) | 0.10 | **0.04** |

[a] the values are only for leaf of understory plants.

Nitrogen addition tended to decrease soil $\delta^{15}N$ values in the BF at all depths, but with no

significant change in each layer as well as across the total soil profile ($p = 0.27$, Fig 2a). In the PF,
N addition tended to decrease soil $\delta^{15}N$ in top 0-10 cm, but the changes were not significant for any
layer or in total soil profile ($p = 0.88$, Fig 2b). When compared based on N pool weighted plot



mean, the two forests did differ significantly in plant N% and $\delta^{15}$N (Fig. 3a). For the soil, the two
forests significantly differ in N pool weighted plot mean N%, with the BF having the higher value,
but not in N pool weighted plot mean $\delta^{15}$N (Fig. 3b).

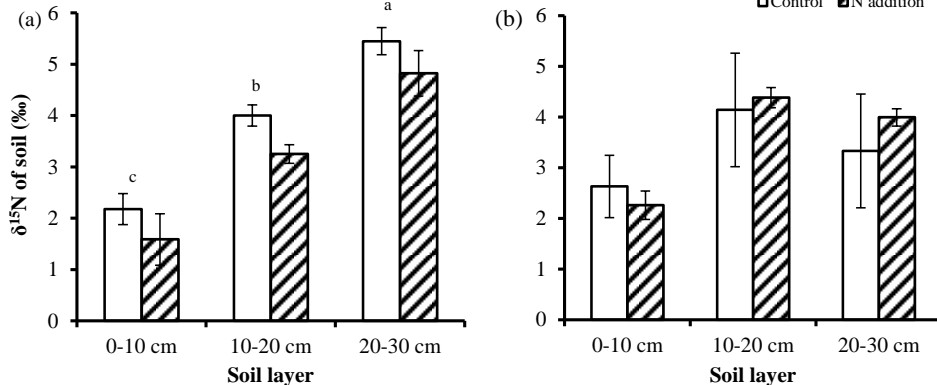


**Fig. 2** Mean $\delta^{15}$N (‰) in the soil profiles of broad-leaved forest (a), and pine forest (b). Error bars indicate
SE of plot means ($n = 3$). Different letters indicate significant difference between soil depths. No significant
difference was found between the two forest types at any depth nor was there any effect of N addition


In summary, the effect of added N on pool weighted plot mean plant N% was neither significant in
BF ($p = 0.86$) nor in PF ($p = 0.2$), but the change was more pronounced in the PF (Fig. 3a).
However, weighted plot mean plant $\delta^{15}$N values were significantly increased in both forests ($p =$
0.03 for BF and 0.01 for PF) from the $^{15}$N-enriched (compared to plant tissues) N addition. In the
soil, where the N pool is obviously larger than in the plants, the effect of the N addition on weighted
average N% was not significant in both forests (Fig. 3b). The direction of change in soil $\delta^{15}$N was a
decrease as expected with incorporation of the added N with a $\delta^{15}$N value of -0.7‰, but the change
was again not significant (Fig. 3b).





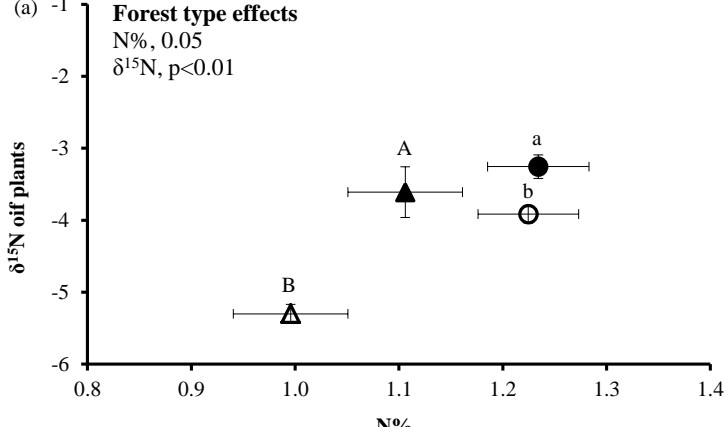


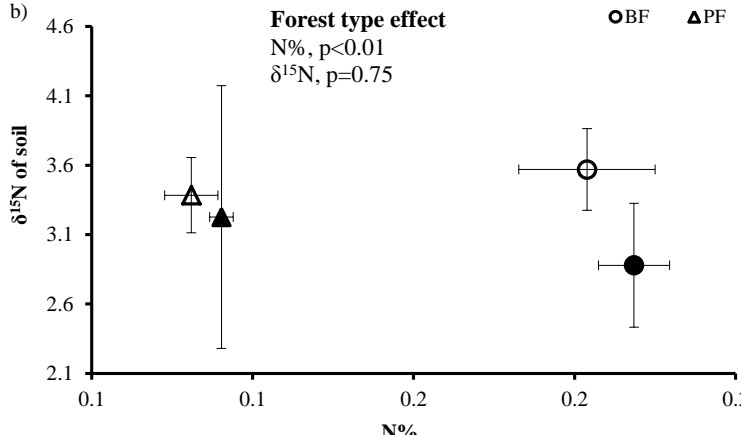


**Fig. 3** Overall effect of N addition on plot average weighted N% and $\delta^{15}$N of plants (a), and soil (b) for broad-leaved forest (○) and pine plantation (△). Error bars indicate SE of plot means ($n = 3$). Open symbols indicate control plots and closed symbols indicate N-plots. In (a), different lowercase letter for BF and uppercase letter for PF in (a) indicate significant difference. The shown $p$-values are tests for differences in N% and $\delta^{15}$N between the two forests (control plots).







### 4. Discussions

#### 4.1. $\delta^{15}N$ of deposition N, plants and soil

The $\delta^{15}N$-depleted deposition N, especially in the $NH_4^+$-N form both in bulk precipitation and throughfall (Table 2) confirms the widely observed results for forests in southern China (Koba et al., 2012; Zhang et al., 2008) and other regions (Freyer 1978; Russel et al., 1998). The $\delta^{15}N$-depleted $NH_4^+$-N could indicate effects of $NH_3$ emissions from agricultural activities, which is usually $^{15}N$-depleted (Bauer et al., 2000). Contrary to $NH_4^+$-N, input $NO_3^+$-N has positive $\delta^{15}N$ again confirming previous reports (Fang et al., 2011a; Koba et al., 2012), and it has been explained as the contribution of $NO_x$ produced by coal combustion. The very low $\delta^{15}N$ of $NH_4^+$-N in the soil solution resemble that in precipitation and throughfall (Table 2), and it is likely due to transport of $^{15}N$-depleted N throughfall through macrospores as supported by the positive relationship between $\delta^{15}N$ of $NH_4^+$-N in soil solution and that in throughfall (Fig. 1b).

The observed leaf $\delta^{15}N$ values ranging from -4 ‰ in BF to -6 ‰ in the PF in DHSBR (Table 4) are within the range (-2 ‰ to -5 ‰) found in eastern Asia (Fang et al., 2011a; Wang et al., 2014; Kitayama and Iwamoto, 2001). However, they are more depleted than the average (3.7 ‰) given for tropical forests in a global synthesis (Martinelli et al., 1999), and those reported from a major survey across Amazonas (3.1 ± 2.3 ‰) by Nardoto et al. (2014). Surprisingly, the average leaf $\delta^{15}N$ are closer to, but even more $^{15}N$-depleted than that of temperate forests (-2.8 ‰) (Martinelli et al., 1999). Several studies (Martinelli et al., 1999; Craine et al., 2009; Craine at al., 2015) suggested that leaf $\delta^{15}N$ increases with N availability. The suggestion was based on the hypothesis that N losses by fractionation pathways at sites with high N availability cause plant and soil to become relatively more $^{15}N$-enriched. Supporting the hypothesis, Craine et al (2009) observed positively correlation between foliar $\delta^{15}N$ and the enrichment factor (difference in $\delta^{15}N$ between leaf and soil), and suggested that it reinforce the idea that the general patterns of N availability can be assessed



with foliar $\delta^{15}$N alone. Our result rejects this hypothesis for the high N deposition systems that we
study because the expected ecosystem enrichment from fractionation occurring during N
transformations under the high N availability at DHSBR is overridden by other depleting factors.
We believe that the main depleting factor at DHSBR (and in other Chinese forests with high N
deposition) is a direct uptake of $^{15}$N-depleted deposition N, particularly $NH_4^+$-N (Table 2) by leaf
from throughfall or by roots from the soil solution (Wang et al., 2014). Sun et al. (2010) found that
tree ring $\delta^{15}$N of *Pinus massoniana* at DHSBR (PF) decreased from 2 ‰ in the 1960s to -1 ‰ in the
late 1990s, and that the decrease was found to coincide with the increasing deposition of $^{15}$N-
depleted N over the last 50 years. This finding also supports our conclusion that the $\delta^{15}$N of the
deposition N sources is more important in explaining patterns of ecosystem $\delta^{15}$N data in highly
polluted regions. Thus, we highlight the importance of region-specific interpretations for patterns of
observed foliar signatures that may not be explained by increased N availability.

Soil $\delta^{15}$N at DHSBR are also below the average for tropical forests, but are closer to that

observed for temperate forests (Martinelli et al., 1999). In agreement with the global trends, plants
at DHSBR are more $\delta^{15}$N-depleted than the soils (Table 4, Fig 2). This trend has been explained by
isotopic discrimination against $^{15}$N during mineralization and nitrification that makes the N
available for plants more $^{15}$N-depleted than the source soil organic matter pools (Amundson et al.,
2003). The increase of soil $\delta^{15}$N with soil depth in the BF is also a well-documented phenomenon in
undisturbed soils (Boeckx et al., 2005; Emmet et al., 1998; Koba et al., 2010). It is caused by
continuous input of $^{15}$N-depleted litter that keeps $\delta^{15}$N of the top soil low, whereas $^{15}$N-enriched soil
fractions are transported and accumulated at deeper soil profile (Nadelhoffer et al., 1988) when the
lighter $^{14}$N is removed by plants, microbes, or through leaching following decomposition (Boeckx
et al., 2005). Enrichment of deeper soil layers could also occur due to fractionation when
mycorrhiza fungi transfer $^{15}$N-depleted N to their host plants $^{15}$N (Högberg et al., 2011; Hobbie and



Högberg, 2012) leading to accumulation of $^{15}$N-enriched N of fungal origin in deeper soil layers.
However, the role of mycorrhiza in N uptake is less in N saturated ecosystems and hence the
increase in $\delta^{15}$N with soil depth may be less steep (Högberg et al., 1996, 2011; Emmett et al., 1998;
Hobbie and Ouimette, 2009). This might explain the minor observed increase in soil $\delta^{15}$N (~3 ‰)
with depth at DHBSR in BF compared to that found for several temperate sites (Vervaet et al.,
2002; Emmet et al., 1998; Bauer et al., 2000), and the 3 to 9 ‰ increase with soil depth reported in
global data from temperate forests (Martinelli et al., 1999).

**4.2. Effects of N addition on N% and $\delta^{15}$N**
The increase in plant $\delta^{15}$N caused by N addition (Table 4, Fig 3a) is consistent with previous
measurements on leaves of the two forests at DHSBR (Fang et al., 2011a) and in several temperate
forests (Högberg et al., 2011; Högberg et al., 2014; Korontzi et al., 2000; McNulty et al., 2005;
Näsholm et al., 1997). It was interpreted mainly as effects of fractionation during N uptake and
cycling as discussed above. For our experiment, we found another explanation more likely: inspite
of the higher N deposition with more negative $\delta^{15}$N than that of plants, plant $\delta^{15}$N moved toward the
more positive $^{15}$N signature of the added N fertilizer (-0.7 ‰), indicating acquisition of the added N
by the plants (Table 4; Fig. 3). However, the changes in $\delta^{15}$N slightly varied among compartments
because they differ in terms of N turnover time. Thus, the most pronounced changes in $\delta^{15}$N
occurred in small but more active pools (e.g. leaves, roots) compared to the large and relatively
inactive/stable pools (e.g. woods and soil pools). Added N is more likely incorporated into the
active pools that are responsive to contemporary N input manipulation (Fang et al., 2006;
Johannisson and Hogberg, 1994; Pardo et al., 2002). Response of weighted plot mean $\delta^{15}$N of the
plants also showed that $\delta^{15}$N of the whole plant pools were significantly increased in both forests
(Fig 3a). The increase in plant $\delta^{15}$N after decadal N addition (Fig 3a), i.e., toward the $^{15}$N signature





of the added N is in line with our second hypothesis, and provides evidence for the incorporation of
the decadal N addition even though the forest is considered as N-rich. A substantial incorporation of
newly added N into the ecosystem N pools was confirmed by a recent tracer study in the BF
(Gurmesa et al., 2016). More importantly, it again showed the importance of $^{15}$N signature of input
N in controlling ecosystem $\delta^{15}$N.

Our result showed an insignificant decrease of soil $\delta^{15}$N by N addition moving the $^{15}$N

signature towards that of the added N (Fig. 3b), again pointing to an imprint of the added N, as also
discussed by (Högberg et al., 2014). This contradicts results in similar long-term experimental N
addition (Högberg, 1991; Högberg et al., 1996, 2011), where an increase in $\delta^{15}$N in N-fertilized
plots also after addition of $^{15}$N-depleted N was observed, and interpreted as fertilizer-induced
fractionation due to increased N transformation rates.

**4.3. Difference between BF and PF**
We observed differences in N% and $\delta^{15}$N that can be interpreted as a difference in N status and N
cycling difference between the two forests. As expected from previous studies, the BF is more N-
rich than the PF as indicated by higher N% in major ecosystem pools in BF (Table 3). Trees leaves
in BF are more $^{15}$N-enriched than the PF (Table 4) as also previously observed by Fang et al.,
(2011c) for a few tree species in the same forests. The two forests also differ in plant species and
successional age, and (Wang et al., 2014) reported that these are important factors that affect $\delta^{15}$N
in different tropical forests in southern China. The difference could be partly related to higher N
cycling rates and subsequent losses of the lighter $^{14}$N in the BF through fractionating processes, and
subsequent plant uptake of $^{15}$N-enriched soil N (Magill et al., 2000; Zhang et al., 2008; Nadelhoffer
and Fry, 1994). On the other hand, leave $\delta^{15}$N in PF can be more affected by $\delta^{15}$N-depleted
deposition as the forest is still expanding in biomass and has lower N availability, thus it might



depend more on the [15]N-depleted atmospheric N input than the BF does. Another explanation could
be that the PF is dominated by *Pinus massoniana*, which has ectomycorrhizal fungi whereas
majority of the plants in the BF have arbuscular mycorrhizal association (Gurmesa, 2016), and
ectomycorrhizal plants are found to be more [15]N-depleted than arbuscular mycorrhizal plants
(Craine et al., 2009; 2015).

No apparent significant difference in soil $\delta^{15}N$ was observed between the BF and PF, except

the clear increase in soil $\delta^{15}N$ with soil depth only in the BF (Fig 2a). Absence of a clear profile
pattern in soil $\delta^{15}N$ with depth in the PF could be due to effects of erosion and soil mixing caused
by human disturbances until recent years. Soil $\delta^{15}N$ are reported to increase with organic matter age
(Bauer et al., 2000), and we expect soil organic matter of the top soil to be older in the BF, because
this layer might have been lost by erosion in the PF as it could be noted from the lower C, N and P
concentration (Table 1), and lack of depth pattern of soil $\delta^{15}N$ in the PF (Fig 2b).

The differences in plant N% and $\delta^{15}N$ between the two forests are also confirmed by their

response to the decadal N addition. In the BF, N addition did not significantly change plant N% in
each plant compartment and in the pool weighted plant pools because the plant tissues was already
saturated with N as also suggested by Fang et a1. (2011c). In contrast, plant N% of tree
compartments (and pool weighed N%) in the relatively N-poor PF tended to increase, and
significantly increased in understory plants (Table 3, Fig. 3a), indicating more dependence of plant
on input N in the PF, although the pool weighed N% was neither significantly increased in the PF.
Although the plant $\delta^{15}N$ after decadal N addition was not significantly different between the two
forests, the more pronounced effect of N addition in PF could still hint an ecological interpretation
of a difference in N status. In line with our expectation, the larger changes in plant $\delta^{15}N$ in the PF
could indicate larger incorporation of added N in to plants in the PF compared to that in the BF. In
support of this, calculations based on an isotope mixing model (Dawson et al., 2002) estimated the



fraction of added N that was incorporated into the plant N pools to be larger (~30 % of the total 500
kg N ha$^{-1}$ over a decade) in PF compared to ~20 % in the BF.

**5. Conclusion**
Our findings show that plants and soil in humid tropical forests of southern China are $^{15}$N-depleted
likely due to an imprint from $^{15}$N-depleted N atmospheric deposition. This effect of the input N
(deposition) $^{15}$N signature was further confirmed by our observation that $\delta^{15}$N of both plants and
soil moved toward the $^{15}$N signature of added fertilizer N, which also shows that the added fertilizer
is incorporated into the forest N pools. We found that broad-leaved forests and early successional
forests differ in their N% and $\delta^{15}$N, and accordingly differ in their response to increased N input.
Significant changes in the overall ecosystem $\delta^{15}$N after decadal N addition in both forests indicate
that the $^{15}$N signature of incoming N is more important for ecosystem $\delta^{15}$N than fractionation during
the steps of N cycling. Thus, the use of ecosystem $\delta^{15}$N as a tool to interpret changes in ecosystem
N cycling suggested in the literature is hampered by the $^{15}$N imprint of increased N deposition,
particularly in regions with high N emissions.

**Acknowledgment**
This study was initiated under the Sino-Danish Centre for Education and Research (SDC), and
supported by the National Basic Research Program of China (2014CB954400), and the National
Natural Science Foundation of China (No. 41473112; 31370498), and the SDC. We would like to
thank the State Key Laboratory of Forest and Soil Ecology, Institute of Applied Ecology, the
Chinese Academy of Sciences (No. LFSE 2013-13) for analyzing the water samples. We also wish
to thank Lijie Deng and Xiaoping Pan for their skillful assistance in field and laboratory work.





**Authors contribution:** Gundersen P. and Mo J. conceived and designed the experiments. Gurmesa
A.G., Lu X., Mao Q. and Zhou K. performed the data acquisition. Gurmesa A.G. analyzed the data.
Gurmesa A.G. and Gundersen P. wrote the manuscript. Lu X. and Mo J. commented and edited the
article.
**Conflicts of interest**: The authors declare that they have no conflict of interest.

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
