# Peer review of "Nitrogen input 15N-signatures are reflected in plant 15N natural"

_Biogeosciences, 2016_

## Referee Comment (RC1) · Anonymous Referee #2 · 21 Dec 2016

Dear authors

The study investigated effects of natural 15N abundance of sources in forest ecosystems on d15N value in two different types of forest ecosystems receiving relatively high nitrogen deposition in China. The study is valuable because there are very few long-term nitrogen addition experiments in the area. The theme of the study is suitable for Biogeosciences. However there are some problems and manuscript should be revised.

Major comments

There are two processes explaining nitrogen isotope ratio; 15N of sources and fractionation processes. Authors discuss the relative contribution of these factors. Authors

concluded that source 15N is more important than fractionation in the study. However, it is very difficult to separate these two processes. Authors stressed the importance of source 15N too much. Description of the manuscript should be revised substantially.

To evaluate the effects of nitrogen addition, nitrogen concentration and d15N values are compered between the control and nitrogen added plots. There only three replication in each treatment and statistical power is very low. Because of this weakness care should be taken when the authors discuss the non-significant results. For example, nitrogen concentration of tree leaves at the pine forest was greater in the N added plot at 10 percent level in table 3. When considering the small number of replication, it is difficult to conclude that there is no significant effect of nitrogen addition. In figure 2, delta 15 N in soil seems different between the control and nitrogen added plots, p-value should be shown for each soil layer and total soil as shown for plant compartments. Authors should describe the limitation of the study about statistical analysis and careful interpretations are required.

Specific comments L 31, leafs ->leaves

L 29-30, d15N value of added nitrogen should be described in the sentence.

L37, "plant N% was unchanged...," nitrogen concentration was marginally increased in pine leaves and significantly in understory vegetation in the pine forest.

L39, "the signal from the input may override...," 'override' is not a proper word in the situation. Fractionation is also an important process for explaining the difference between plant and soil and between soil depths.

L137, duration of nitrogen addition should be clearly shown.

L192-193, information of surface runoff is not sufficient. What is the size of the barrier? How did you collect the water samples?

L229-230, p-value of statistical analysis should be shown in table 2.

L235, 'see page' should be 'soil solution.' 15N value of total inorganic N (NH4 plus NO3) should be helpful.

L238, section title should be revised. It would be "effects of forest types."

L239-240, the information about earlier study should be described in discussion.

L240-241, clearly indicate that this comparison is about the control plots.

L245-247, p-value should be shown. Information about fig 2 should be included in table 4 as shown for table 3.

L263, p-value of statistical analysis should be described.

L295-299, information of the graphs should be included in table 4 and statistical analysis for N addition should be shown.

L312-316 Figure 3, Information of the graphs should be presented in tables as shown in table 3 and 4. The effects of nitrogen addition should be indicated. Nitrogen concentration and d15N of whole ecosystem (plant plus soil) would be helpful.

L329-332, Mean value of d15N of soil solution is much lower than throughfall or precipitation. Is there any reason for this difference?

L345-347, when you compare the d15N value between BF and PF, pine forest had lower d15N. The positive correlation between N availability and leaf 15N still exists within the area. Therefore, it is difficult to conclude the results reject the hypothesis.

L362-375, description is only about BF. Is there any comment on PF?

L382, the contribution of fractionation process and source 15N value is not clearly known in this study.

L392-397, it is difficult to understand. N addition possibly decreases the fractionation during n mineralization and may increase plant 15N. It is difficult to conclude that 15N source is main sole factor.

L398-403, the results are based on non-significant results. It is very difficult to conclude the decrease is due to 15N of added N. Because N input by thoughfall has lower value than added N, 15N of total N input should be lower in the control plots. I thought the description is not correct.

L405-442, the section should be moved to just before the previous section 4.2.

L415-416, it is difficult to conclude that source 15N is more important in PF. It is too speculative.

L436-439, description about N addition should not be described in this section.

L445-455, it is difficult to conclude that 15N of source is more important than fractionation process. The contribution of fractionation is still also important factor. Conclusion should be revised substantially.

———————————————————

---

## Referee Comment (RC2) · W. W. Wessel (Referee) · 27 Dec 2016

General comments

This paper discusses two processes that affect the delta 15N of forests. Firstly the mixing with N deposition with a different 15N signature than the forest itself and secondly the fractionation of the 15N through different N transformation processes followed by the loss of the lighter fraction resulting in enrichment of the remaining N with 15N. The latter process is thought to happen more strongly if N availability is larger and so it is thought that a higher delta 15N is an indication of a higher N availability.

The authors present two sets of delta 15N results of two forests in southern China:

the results of the ambient situation ('control') and the results of a long term N addition experiment in the same forests. In the control experiment they find rather low delta 15N values compared to literature values. As the delta 15N value of the substantial N deposition is also rather low, they conclude that the mixing with N deposition with a low delta 15N is the dominating process in determining the 15N of the vegetation and fractionation combined with loss is not important. Secondly they discuss the effects of a long term addition of N with a higher delta 15N than that of the ambient deposition. Here they conclude that the increase in delta 15N in the vegetation is not the result of increased fractionation and loss, due to the higher N availability but that this is the result of the mixing of the added N with a high delta. Their general conclusion is that when delta 15N of forests is used to say something about N availability more attention should be given to the possible influence the delta 15N signature of the N deposition can have.

Although I do not think that the main conclusions of the authors are incorrect, I think their argument does need substantial improvement.

1. In the first place they do not make clear what delta 15N value they do expect for their forest (under ambient conditions) as a result of fractionation and loss, as described in their hypothesis i (line 101). This hypothesis i is unclear (see below) and they do not explicitly compare this hypothesis with their results. In the Discussion section they compare their results not with individual forests from the literature but with large datasets synthesized from many different forests. Why would their conclusions about their own forests not be true for the forests they cite from the literature? If not, what could be the relevant differences between their forests and those from the literature? Maybe the values calculated for southern China by Amundson et al (2003), based on MAP and MAT can help to structure this part of the discussion? 2. At first sight it seems reasonable to consider mixing to be important in the control experiment, but this could be supported with some calculations of the effect of mixing. It seems the authors have carried out such calculations at least for some cases according to their statement in line 440, but

this would be useful for this case as well. 3. The reasoning in lines 339-356 is very difficult to follow. I will make more specific comments below.

4. Concerning the N addition experiment it can be said that both the mixing process and the increased fractionation plus loss process (expected as a result of larger N availability) would lead to an increase in the delta 15N of the vegetation, so it is unclear why the authors choose that the increased delta 15N values found in the vegetation were the result of mixing and not of increased fractionation plus loss. What result of the experiment and the measurements would have led them to the other conclusion? In fact probably both mixing and fractionation plus loss contribute to some extent to the increase of delta 15N in the vegetation. Again some calculations of the mixing of the deposition might give more insight into the potential contribution of this process.

5. I think there is something wrong with the statistical results presented in Tables 3 and 4. The tests for siginifant differences sometimes yield significant p values while the difference tested is smaller than the sum of the two standard errors. This cannot be correct. I suggest the authors provide the data and the script they have used to calculate the statistics so it becomes clear what they have done. See for example in Table 3 twigs difference between BF and PF is 0.29, while the sum of the SEs is 0.96 and p<0.01 and in Table 4 tree leaf in BF difference between control and N addition is 0.6, while the sum of the SEs is 1.1 and p<0.01. I assume two-sided tests were carried out although this was not mentioned.

6. I would suggest that the authors should be more careful in using the terms 15N enriched and 15N depleted and define what exactly is meant by them and relative to what (below or above zero, or relative to the delta of some other pool or flux). They use these terms many times throughout the text. See e.g. my comment below on line 402. In line 32 even the term "more enriched" is used.

Specific comments

L25 "examined the measurement": this suggests the paper is about measurement

techniques. I suggest to rephrase this.

L31 "leafs" the text contains many spelling errors; I suggest the authors check the text throughout for these.

L31: "old-growth forest" this forest is everywhere else described as broadleaved forest, so I would suggest to use that term here as well

L48 "recently" I think it is relevant to be more specific, so the reader knows how long this N addition has been going on. In the methods the 1990s are mentioned for DHSBR (L113).

L67 "above the atmospheric standard" I wonder whether for this criterion 0.0 0/00 is the relevant value, as atmospheric N2 is not a direct source of N for a terrestrial ecosystem.

L81-82 "hotspots" If this is meant to be high in N deposition, I would suggest to use the latter term.

L102 The comparison of a high N forest with temperate forests seems inconsequent. What about temperate forests with high N status?

L103 second hypothesis: I wonder which results could lead to the rejection of this hypothesis, given the experimental conditions. The first part of this hypothesis seems not very challenging and the second part is not very specific.

L114 "steep slopes" Amundson et al (2003) have suggested that under these circumstances delta 15N might be lower (see their paragraph [26])

L182 "including a dry period" If the authors mean that there were not any water samples in Dec and Jan because of a lack of precipitation then please state this.

L186-187 A collector with an area of 8000 cm2 seems extremely large. Is this a correct value?

L215 "plant species as a random factor" Apparently this is not the case for the pine

forest, which contains only one dominant species (L159).

L233 Table 2 At first sight it looks like leaching losses have lower deltas than the deposition, indicating the occurrence of fractionation and loss of N with low deltas, thus increasing the 15N content of the remaining N. However, as deposition is dominated by NH4, while leaching is dominated by NO3, this is not the case. Calculating a weighted average delta 15N for all chemical species in all fluxes may show this. This can support the argument that fractionation plus loss is not evident from this budget, although it is of course incomplete. Are there not any values for the added N plots?

L233 Table 2 In the text it is stated that runoff was measured only in one plot per treatment (line 192), so how can there be an average of three measurments for runoff here?

L251 Fig.1 Are these samples that were taken monthly between Sept and Feb (4 samples) with 3 replicates, in total 12? I suggest to explain this in the caption. Again how was this for runoff (see my previous remark on Table 2)? What could be the cause for the variation found? Is there no substantial time delay between the moment of deposition and the moment the deposited N reaches the subsoil or the runoff?

L261-263 "N concentration of N pool weighted average plant pools calculated per plot". The reader is referred to Table 3, but in there are only N concentrations of individual pools.

L295 Fig.2 I suggest to increase the size of the symbols in the legend so the different patterns used are more easily recognized. This is also a problem in the supplement figure.

L307 "decrease as expected" It is true that the delta of the N input into the forest is still lower than the the delta of the soil, but the addition has substantially increased the delta of the total N input, so one might as well expect an increase in the soil delta as a result of this.

L325 "other regions" please specify which regions are meant.

L337 "surprisingly" I suggest the authors clarify what they expect here.

L342-345 This remark on the enrichment factor seems misplaced here, as nowhere else in the paper something is said about the enrichment factor. It is also unclear to me why this would support the previously mentioned hypothesis.

L345 "rejects this hypothesis" Which result precisely makes the authors decide to reject? Do the authors reject the full hypothesis or only mean that the increase in delta 15N simply does not happen? Nothing is said about hypothesis i from the introduction. I would suggest to refer to this hypothesis as well, although it needs to be rephrased, as I mentioned earlier.

L347 "other depleting factors" I think "other" should be removed as the previously mentioned process is an enriching factor.

L348-349 "in other Chinese forests with high N deposition" Why only or especially in Chinese forests? And would this not depend on the delta 15N value of the N deposition? Maybe the authors have the literature in mind they mention in lines 323Â■ . If that is the case they should refer explicitly to these results. The authors make a different and more general statement in lines 454-455.

L381 "It was interpreted" I suppose this was done by the references mentioned just before this sentence. To make this clearer to the reader I suggest to change the sentence from passive into active voice.

L393 "in line with our second hypothesis" This can only be true for the first part of this hypothesis

L396-397 "it shows again" I disagree. From these results one could argue as well that it is the result of increased N availability resulting in increased fractionation plus loss of depleted N.

L402 "also after addition of 15N depleted N" In the experiment by the authors the N added was 15N enriched (at least compared to the ambient N deposition).

L440 "calculations based on an isotope mixing model" I would suggest to add some information on how this was calculated and which simplifying assumptions were made in the calculation.

L445 "in humid tropical forests of southern China" why would this be true for all these forests, not just for the forest investigated? Possibly because of the delta 15N value of the deposition there (see line 323‑ )? Then the authors should refer to this. Would the region differ in this respect from other regions in the world?

L447 "further confirmed" see my remark on L396

L452 "more important" this is only the case if the 15N signature of the Ndeposition differs sufficiently from the delta 15N of the ecosystem, and the N deposition is sufficiently large. If that is not the case the mixing probably would not dominate the fractionation plus loss of depleted N.

---

## Author Comment (AC1) · 24 Feb 2017

Response to Anonymous Referee #2

The study investigated effects of natural 15N abundance of sources in forest ecosystems on d15N value in two different types of forest ecosystems receiving relatively high nitrogen deposition in China. The study is valuable because there are very few long term nitrogen addition experiments in the area. The theme of the study is suitable for Biogeosciences. However there are some problems and manuscript should be revised. Response: Thanks for the constructive comments and suggestions that were very useful to improve our manuscript. We have revised our manuscript by implementing those suggested changes and/or by adding more explanation to clarify our points.

Major comments

There are two processes explaining nitrogen isotope ratio; 15N of sources and fractionation processes. Authors discuss the relative contribution of these factors. Authors concluded that source 15N is more important than fractionation in the study. However, it is very difficult to separate these two processes. Authors stressed the importance of source 15N too much. Description of the manuscript should be revised substantially.

Response: We highlight the importance of 15N signature of sources because of three reasons: First, plants are more 15N-depleted at our study site than seen in global data for tropical forests (Martinelli et al. 1999 and others cited in the manuscript (e.g., Nardoto et al. (2014). According to these studies tropical forests are 15N-enriched because of increased N cycling rates due to high N availability. Since we found that deposition N is not only high, but also strongly 15N-depleted, we argued that 15N of sources add an imprint on top of the effects from fractionation processes. Second, we found that experimental addition of N (with delta 15N different from that of the plants) moved the delta 15N of ecosystem pools (plant and soil) toward the 15N signature of the added N, pointing to the importance of 15N of sources for ecosystem delta 15N. Third, when we used the added N with -0.7 delta as tracer for mass balance calculation (Dawson et al. 2002), about 20% of the added N was estimated to be taken up by the plants in the BF. This was close to the estimated fate (24% to plants) of a stronger tracer (Gurmesa et al. 2016) and thus hint that the input N is substantially incorporated into plants although they over all do not increase the uptake in BF. Finally, a recent paper (Perakis et al. 2015) also highlighted the importance of the 15N of source N, although it was from N-fixation. We agree that it is difficult to separate the contribution of the two processes. To address this concern, we have revisited our text and conclusions to moderate and clarify the statements on the effect of 15N sources, not to oversell the point. Probably our use of the word 'override' is part of overstating the case. In the revision we use 'dominates effects of fractionation' and keep mentioning also the fractionation signal.

To evaluate the effects of nitrogen addition, nitrogen concentration and d15N values are compered between the control and nitrogen added plots. There only three replication in each treatment and statistical power is very low. Because of this weakness care should be taken when the authors discuss the non-significant results. For example, nitrogen concentration of tree leaves at the pine forest was greater in the N added plot at 10 percent level in table 3. When considering the small number of replication, it is difficult to conclude that there is no significant effect of nitrogen addition.

Response: Lack of enough replication is the common limitation of N addition as well as isotope studies. Not many studies have used three true replicate plots for this kind of studies in tropical forests. We agree that the statistics analysis is not very strong to make strong conclusion. Note however, that for the broadleaf forest 5 dominant tree species were sampled and since the species differ in %N and delta 15N species was included as a random factor in the tests; i.e. for plant compartments the N addition effects build on more than just three determinations. We have checked our wording in the results section carefully to avoid such weakly supported statements.

In figure 2, delta 15 N in soil seems different between the control and nitrogen added plots, p-value should be shown for each soil layer and total soil as shown for plant compartments. Authors should describe the limitation of the study about statistical analysis and careful interpretations are required.

Response: We agree to add the data in Fig 2 into Table 4 so that the p-values asked for can be added and shown in the same way as for plants.

Specific comments

L 31, leafs ->leaves

Response: Done

L 29-30, d15N value of added nitrogen should be described in the sentence.

Response: We have added that the delta 15N of the added N is close to that of atmospheric N

L37, "plant N% was unchanged. . .," nitrogen concentration was marginally increased in pine leaves and significantly in understory vegetation in the pine forest.

Response: We have revised the sentence, indicating the directions (tendency) of changes though it is not significant. We agree that the term 'unchanged' may not be appropriate for the said reason.

L39, "the signal from the input may override," 'override' is not a proper word in the situation. Fractionation is also an important process for explaining the difference between plant and soil and between soil depths.

Response: In the revision we use 'dominates the signal' instead of 'override' and keep mentioning also the fractionation effects (see our response above).

L137, duration of nitrogen addition should be clearly shown.

Response: We have mentioned that the N addition experiment was established in July 2003, and that it is ongoing now. We have also mentioned that the N is added monthly.

L192-193, information of surface runoff is not sufficient. What is the size of the barrier? How did you collect the water samples?

Response: We have provided more information on how we sampled the surface runoff using the steep slope nature of the plots.

L229-230, p-value of statistical analysis should be shown in table 2.

Response: No significant difference was detected between the two forests. This information is now included in the table caption.

L235, 'see page' should be 'soil solution.' 15N value of total inorganic N (NH4 plus NO3) should be helpful.

Response: The change is made as suggested. '15N value of total inorganic N (NH4

plus NO3)' is also provided.

L238, section title should be revised. It would be "effects of forest types."

Response: Revised as suggested.

L239-240, the information about earlier study should be described in discussion.

Response : Mention of earlier study moved to the discussion.

L240-241, clearly indicate that this comparison is about the control plots.

Response: Now we have clearly indicated that this comparison is about the control plots in the discussion.

L245-247, p-value should be shown. Information about fig 2 should be included in table 4 as shown for table 3.

Response: The data in Fig.2 is now included into Table 4 as suggested.

L263, p-value of statistical analysis should be described.

Response: The p-values were already given, but at another place. Hence, we deleted this part.

L295-299, information of the graphs should be included in table 4 and statistical analysis for N addition should be shown.

Response: Changes implemented as suggested.

L312-316 Figure 3, Information of the graphs should be presented in tables as shown in table 3 and 4. The effects of nitrogen addition should be indicated. Nitrogen concentration and d15N of whole ecosystem (plant plus soil) would be helpful.

Response: As suggested we added the data to the tables as well. We did not give the combined plant+soil values because i) the soil pool is large and will dominate; ii) the direction of change in delta 15N due to N addition go in opposite direction because the

N addition of almost zero delta 15N (of the added N) is right between the level in plants and soil.

L329-332, Mean value of d15N of soil solution is much lower than throughfall or precipitation. Is there any reason for this difference?

Response: Thanks for raising this issue. We have added a discussion of the fractionation pathways including i) nitrification of deposition NH4, that should deplete NO3 product further to be highly negative; ii) nitrification of soil N, where NO3 would still be depleted, but coming from delta 2-5, would dilute the negative signal from i); and denitrification of NO3 to N2 that would enrich the remaining NO3. Since high rates of denitrification have be measured at the site 30 kg/ha/yr (Fang et al. 2015) this may explain why NO3 in soil solution is only slightly depleted. L345-347, when you compare the d15N value between BF and PF, pine forest had lower d15N. The positive correlation between N availability and leaf 15N still exists within the area. Therefore, it is difficult to conclude the results reject the hypothesis. Response: The reviewer has raised good point here. We want to emphasis that the overall ecosystem 15N-enrichment in tropical ecosystems due to increased N cycling rate is absent at our study site. Here we compared the 15N at DHSBR to other tropical forests reviewed by several authors (e.g., Martinelli et al., 1999; Craine et al., 2009; Craine at al., 2015). We have clarified our wording here, including the use of the word 'overridden' as mentioned above. We acknowledge the difference between the two forests that support the hypothesis because the N-rich BF forests are more 15N-enriched than the relatively N-limited PF.

L362-375, description is only about BF. Is there any comment on PF?

Response: We did not focus on the PF because we observed no clear patter in soil delta 15N with soil depth. We believed this to be due to the effects of erosion and soil mixing caused by human disturbances until recent years in the PF. This was explained under section 4.2 (previous section). Now we have explained this under this section to

address the concern by the reviewer.

L382, the contribution of fractionation process and source 15N value is not clearly known in this study.

Response: We agree that the contribution of fractionation process and 15N signature of source cannot be separated using the current data. The statement was not meant to rule out fractionation but to show that the 15N signature of source N can explain the change. Again we changed the wording.

L392-397, it is difficult to understand. N addition possibly decreases the fractionation during n mineralization and may increase plant 15N. It is difficult to conclude that 15N source is main sole factor.

Response: We added more text here to explain the figure; hope it helped. The globally established knowledge is that increased N input increases fractionation, and we do not have evidence that N addition might have reduced fractionation during mineralization. However, we agree that fractionation can still be important contributing factor and we had no intention to boldly say that '15N source is main sole factor'. We have revised the discussion to elaborate this.

L398-403, the results are based on non-significant results. It is very difficult to conclude the decrease is due to 15N of added N. Because N input by thoughfall has lower value than added N, 15N of total N input should be lower in the control plots. I thought the description is not correct.

Response: We agree that '15N of total N input should be lower in the control plots'. We assume that delta 15N of thoughfall equally affects plants in control and N-plots equally. Any difference between in delta 15N is likely due to the effect of the N addition. Obviously the effect of N addition was not significant due the large soil N pool that is less responsive to contemporary N input manipulation. But, the observed trend supports or explanation on the importance of 15N signature source input.

L405-442, the section should be moved to just before the previous section 4.2.

Response: We have moved the section previous section 4.2.

L415-416, it is difficult to conclude that source 15N is more important in PF. It is too speculative.

Response: We have indication that incorporation of the added N is high in the PF. Change in plant N content was higher in the PF, indicating the input N has more effect on plants in the PF than in BF. Based on these observations, we argued that input N is more important for plant N source in the relatively N-limited PF. Our conclusion on the response of delta 15N is correct. We have revised the sentence to make it not too speculative that is not supported by data. We connected our statement to the result of the isotope mixing calculations that were further down in line 440-442.

L436-439, description about N addition should not be described in this section.

Response: We were not sure what the reviewer meant here. But this part of the text was changed also in response to the other reviewer.

L445-455, it is difficult to conclude that 15N of source is more important than fractionation process. The contribution of fractionation is still also important factor. Conclusion should be revised substantially.

Response: We have revised the conclusion with careful wording as also detailed in our response to the general comments above.

---

## Author Comment (AC2) · 24 Feb 2017

Response to W. W. Wessel (Referee)

General comments

This paper discusses two processes that affect the delta 15N of forests. Firstly the mixing with N deposition with a different 15N signature than the forest itself and secondly the fractionation of the 15N through different N transformation processes followed by the loss of the lighter fraction resulting in enrichment of the remaining N with 15N. The latter process is thought to happen more strongly if N availability is larger and so it is thought that a higher delta 15N is an indication of a higher N availability.

The authors present two sets of delta 15N results of two forests in southern China: the results of the ambient situation ('control') and the results of a long term N addition experiment in the same forests. In the control experiment they find rather low delta 15N values compared to literature values. As the delta 15N value of the substantial N deposition is also rather low, they conclude that the mixing with N deposition with a low delta 15N is the dominating process in determining the 15N of the vegetation and fractionation combined with loss is not important. Secondly they discuss the effects of a long term addition of N with a higher delta 15N than that of the ambient deposition. Here they conclude that the increase in delta 15N in the vegetation is not the result of increased fractionation and loss, due to the higher N availability but that this is the result of the mixing of the added N with a high delta. Their general conclusion is that when delta 15N of forests is used to say something about N availability more attention should be given to the possible influence the delta 15N signature of the N deposition can have.

Although I do not think that the main conclusions of the authors are incorrect, I think their argument does need substantial improvement.

Response: Thanks for the constructive comments and suggestions. We have seriously addressed all the concerns. For each general and specific comment, we have provided our responses, indicating all the changes we made to the manuscript.

1. In the first place they do not make clear what delta 15N value they do expect for their forest (under ambient conditions) as a result of fractionation and loss, as described in their hypothesis i (line 101). This hypothesis i is unclear (see below) and they do not explicitly compare this hypothesis with their results. In the Discussion section they compare their results not with individual forests from the literature but with large datasets synthesized from many different forests. Why would their conclusions about their own forests not be true for the forests they cite from the literature? If not, what could be the relevant differences between their forests and those from the literature? Maybe the values calculated for southern China by Amundson et al (2003), based on MAP and

MAT can help to structure this part of the discussion?

Response: Thank you for the suggestion for using the Amundsen study to initiate the discussion part. We have strived to clarify our hypotheses and their discussion along the lines suggested. Now we have clearly indicated that we expect delta 15N value under ambient conditions to be higher for our study forest in our re-phrased hypothesis. We have also compared the hypothesis (all hypotheses) with our results. The discussion is improved by comparing our results to both individual forests from the literature (temperate forest which we missed in previous version) and large datasets synthesized from many different forests across the world.

2. At first sight it seems reasonable to consider mixing to be important in the control experiment, but this could be supported with some calculations of the effect of mixing. It seems the authors have carried out such calculations at least for some cases according to their statement in line 440, but this would be useful for this case as well.

Response: As suggested we did further calculations of the effect of mixing in the control plot using two-source mixing model (Dawson et al. 2002, cited in the manuscript), where we assumed soil and deposition N as the major N source to plants. The result showed fraction of N contributed by deposition is 60-80% in the two forests with the higher value being for the pine forest. If plants uptake such large proportion of the deposited N, it likely influence delta 15N of plants as we explained in the discussion because deposition is high at our site and it is strongly 15N-depleted. See further details below.

3. The reasoning in lines 339-356 is very difficult to follow. I will make more specific comments below.

Response: We have made substantial changes to the texts to make it clearer. For example, instead of comparing our result only to the global data in Martinelli et al. 1999, we have added more citation from individual studies from temperate forests that include data from temperate sites that have high N status. We also made it clear how

our original first hypothesis was rejected by our results. See our responses to the specific comments too.

4. Concerning the N addition experiment it can be said that both the mixing process and the increased fractionation plus loss process (expected as a result of larger N availability) would lead to an increase in the delta 15N of the vegetation, so it is unclear why the authors choose that the increased delta 15N values found in the vegetation were the result of mixing and not of increased fractionation plus loss. What result of the experiment and the measurements would have led them to the other conclusion? In fact probably both mixing and fractionation plus loss contribute to some extent to the increase of delta 15N in the vegetation. Again some calculations of the mixing of the deposition might give more insight into the potential contribution of this process.

Response: We agree that fractionation combined with loss of 15N-depleted N forms can be caused by larger N availability, and it can be one of those important factors that control ecosystem delta 15N. The increase in plant delta 15N after decadal N addition can be partly explained by this fractionation plus loss processes. However, the same fractionation may not explain the tendency of decrease in soil delta 15N. When we used the added N with -0.7 delta label as tracer for mass balance calculation (Dawson et al. 2002) about 20% of the added N was estimated to be taken up by the plants in the BF. This was close to the estimated fate (24% to plants) of a stronger tracer (Gurmesa et al. 2016) and thus hint that the input N is substantially incorporated into plants although they over all do not increase the uptake in BF. See our response to similar question from the other referee.

5. I think there is something wrong with the statistical results presented in Tables 3 and 4. The tests for siginifant differences sometimes yield significant p values while the difference tested is smaller than the sum of the two standard errors. This cannot be correct. I suggest the authors provide the data and the script they have used to calculate the statistics so it becomes clear what they have done. See for example in Table 3 twigs difference between BF and PF is 0.29, while the sum of the SEs is 0.96

and p<0.01 and in Table 4 tree leaf in BF difference between control and N addition is 0.6, while the sum of the SEs is 1.1 and p<0.01. I assume two-sided tests were carried out although this was not mentioned.

Response: For the comparison of delta 15N between the two forests, we have mentioned that t-test was used as the reviewer assumed it. In Table 3, we had made a copy-past mistake; the correct SE for twigs in PF is 0.05, not 0.77. Further, it is important to note (as mentioned in the table heading) that for the broadleaf forest 5 dominant tree species were sampled and since the species differ in %N and delta 15N, species was included as a random factor in the tests using mixed ANOVA (mentioned in section 2.5); i.e. for plant compartments the N addition effects build on more than just three measurements. Thus the overlap of the SE's based plot means may not be instructive in that case.

6. I would suggest that the authors should be more careful in using the terms 15N enriched and 15N depleted and define what exactly is meant by them and relative to what (below or above zero, or relative to the delta of some other pool or flux). They use these terms many times throughout the text. See e.g. my comment below on line 402. In line 32 even the term "more enriched" is used.

Response: We have strived to clarify the wording and have in some cased added the delta changes to improve on this.

Specific comments

L25 "examined the measurement": this suggests the paper is about measurement techniques. I suggest to rephrase this.

Response: We have rephrased the sentence

L31 "leafs" the text contains many spelling errors; I suggest the authors check the text throughout for these.

Response: We have changed 'leafs' to 'leaves'. Similar spelling errors were thoroughly

searched for and corrected.

L31: "old-growth forest" this forest is everywhere else described as broadleaved forest, so I would suggest to use that term here as well

Response: Corrected as suggested

L48 "recently" I think it is relevant to be more specific, so the reader knows how long this N addition has been going on. In the methods the 1990s are mentioned for DHSBR (L113).

Response: We have made it more specific, and mentioned that increase in N deposition in China has been increasing continuously since the 1980s (Liu et al. 2011).

L67 "above the atmospheric standard" I wonder whether for this criterion 0.0 0/00 is the relevant value, as atmospheric N2 is not a direct source of N for a terrestrial ecosystem.

Response: We have deleted the 'above the atmospheric standard' because the atmospheric N may not be direct source of N to the plants.

L81-82 "hotspots" If this is meant to be high in N deposition, I would suggest to use the latter term.

Response: We have deleted 'hotspots' and used 'high N deposition'

L102 The comparison of a high N forest with temperate forests seems inconsequent. What about temperate forests with high N status?

Response: We have re-phrased our hypothesis to indicate that the comparison include temperate forests in general (not only N-limited forests. Similarly, the discussion is improved by adding some studies from different temperate forests that cover wider gradient of N availability including N-saturated forests (Koopmans et al. 1997).

L103 second hypothesis: I wonder which results could lead to the rejection of this hypothesis, given the experimental conditions. The first part of this hypothesis seems

not very challenging and the second part is not very specific.

Response: We have separated the hypothesis into two (see our response to your comment on 393). We have referred each of our three hypotheses in the discussion and explained if they are confirmed or rejected by our results.

L114 "steep slopes" Amundson et al (2003) have suggested that under these circumstances delta 15N might be lower (see their paragraph [26])

Response: We are aware that topography can have a significant influence on the landscape-scale patterns of plant and soil $\delta$15N. The preliminary data set presented in Amundson et al (2003) was from the central California coast range (Figure 4), suggested that up to 2% variation in soil $\delta$15N at a given location. Apart from these observation used to explain within site variation in soil $\delta$15N, we found no study that compared sites with different topography. We do not believe this steep slope at DHSBR is important factor to explain the distinct 15N-depletion compared to other tropical forests. But we have mentioned this as a potential (minor) influence in the discussion.

L182 "including a dry period" If the authors mean that there were not any water samples in Dec and Jan because of a lack of precipitation then please state this.

Response: We have indicated that these dry months are the period when there were not any water samples because of a lack of precipitation.

L186-187 A collector with an area of 8000 cm2 seems extremely large. Is this a correct value?

Response: The 0.8m2 was a total interception area for the five collectors. We have corrected it.

L215 "plant species as a random factor" Apparently this is not the case for the pine forest, which contains only one dominant species (L159).

Response: Mixed model ANOVA was where plant species used as a random factor was

used for compartments with mean from several species (canopy tree in BF and under-story vegetation in both forests. For canopy layer in PF, only one dominant species, a simple t-test was used. We have clarified this in the manuscript.

L233 Table 2 . At first sight it looks like leaching losses have lower deltas than the deposition, indicating the occurrence of fractionation and loss of N with low deltas, thus increasing the 15N content of the remaining N. However, as deposition is dominated by NH4, while leaching is dominated by NO3, this is not the case. Calculating a weighted average delta 15N for all chemical species in all fluxes may show this. This can support the argument that fractionation plus loss is not evident from this budget, although it is of course incomplete. Are here not any values for the added N plots?

Response: Using delta 15N data in Table 2 and the concentration data presented in Fig. S1, we calculated the weighted average delta 15N for total dissolved inorganic N (DIN). The data (presented in revised Table 3) showed that soil solution ass slightly higher deltas 15N than the deposition in both forests. We have two samples for the added N plots but only in the broadleaved forest (BF), and it shows even higher delta 15N∼ -2.8‰ the values in the control plot (-5.7‰ Table 2). Since nitrate is enriched in soil solution, it may appear to show no evidence for fractionation and loss of N with low deltas. However, this argument may not be true if denitrification (reported at our site) dominates nitrification because the two processes have antagonistic effect on delta 15N of the nitrate. The nitrate measured in the soil solution also comes from nitrification of soil ammonium, which can still be enriched compared to the nitrate in deposition N. These points are now added to the discussion about delta 15N of the water samples (also see our response to comments on L329-332 by the other referee).

L233 Table 2 In the text it is stated that runoff was measured only in one plot per treatment (line 192), so how can there be an average of three measurements for runoff here?

Response: The runoff was collected in one plot per treatment, but at three different

points, which we used as replicate. Since these are not true replicates, we have removed the SE and sample size.

L251 Fig.1 Are these samples that were taken monthly between Sept and Feb (4 samples) with 3 replicates, in total 12? I suggest to explain this in the caption. Again how was this for runoff (see my previous remark on Table 2)? What could be the cause for the variation found? Is there no substantial time delay between the moment of deposition and the moment the deposited N reaches the subsoil or the runoff?

Response: Yes, these are samples that were taken monthly between Sept and Feb (4 samples) with 3 replicates, making total of 12 samples. For the surface runoff, sampling was done only in one plots, but at three points, which were shown in the graph. We have explained this in the caption.

L261-263 "N concentration of N pool weighted average plant pools calculated per plot". The reader is referred to Table 3, but in there are only N concentrations of individual pools.

Response: This sentence is now deleted because it was repeated. Data on effects of N addition on N pool weighted average plant pools was correctly presented later and appropriately referred to Fig 2 (which was Fig 3 in the previous version).

L295 Fig.2 I suggest to increase the size of the symbols in the legend so the different patterns used are more easily recognized. This is also a problem in the supplement figure.

Response: Information in Fig 2 is now included in Table 4 based on a comment from another reviewer. We have increased the size of the symbols in all other figurers including the supplement figure.

L307 "decrease as expected" It is true that the delta of the N input into the forest is still lower than the delta of the soil, but the addition has substantially increased the delta of the total N input, so one might as well expect an increase in the soil delta as a result of
this.

Response: Our expectation was that added N has lower delta 15N than the soil, and its incorporation into the soil N pools would lower the soil delta 15N. The only way the added N can increase soil delta is if it caused substantial fractionation that results in loss of 15N-depleted N form and enrichment of soil N pool. This was not proved to be the case at our study site.

L325 "other regions" please specify which regions are meant.

Response: Other regions mean those in Germany (Freyer 1978) and Chesapeake Bay (Russel et al., 1998). However, we have no evidence for 15N-depleted deposition N in these regions, and the two cited papers are also old studies. So we have deleted this part of the sentence and explained our finding in relation to other relevant studies.

L337 "surprisingly" I suggest the authors clarify what they expect here.

Response: Our expectation, as stated in our first hypothesis, was enrichment of leaf $\delta$15N at our study cite that is higher than the average leaf $\delta$15N observed for temperate forests on global scale.

L342-345 This remark on the enrichment factor seems misplaced here, as nowhere else in the paper something is said about the enrichment factor. It is also unclear to me why this would support the previously mentioned hypothesis.

Response: We agree with this point, and have deleted the sentence.

L345 "rejects this hypothesis" Which result precisely makes the authors decide to reject? Do the authors reject the full hypothesis or only mean that the increase in delta 15N simply does not happen? Nothing is said about hypothesis i from the introduction. I would suggest to refer to this hypothesis as well, although it needs to be rephrased, as I mentioned earlier.

Response: The hypothesis was that tropical forests that have high soil N availability due

to increased N deposition have higher delta 15N compared pre-dominantly N-limited temperate forests due to increased fractionation combined with loss of 15N-depleted N in tropical forests. Since we did not observe such ecosystem enrichment at our site, which also has high N deposition, we concluded that the hypothesis is rejected. To make it clearer, we have re-phrased the whole sentence, and indicated why our result rejects this hypothesis. We have also re-structured the paragraphs, and re-phrased the hypothesis to make it brief and direct.

L347 "other depleting factors" I think "other" should be removed as the previously mentioned process is an enriching factor.

Response: The sentence is re-phrased in connection with our response to the above comment on line 345.

L348-349 "in other Chinese forests with high N deposition" Why only or especially in Chinese forests? And would this not depend on the delta 15N value of the N deposition? Maybe the authors have the literature in mind they mention in lines 323Â . If that is the case they should refer explicitly to these results. The authors make a different and more general statement in lines 454-455.

Response: Yes, we refer to those for which we already cited references (Fang et al., 2011a; Wang et al., 2014; line 334 not 323). We have explicitly clarified this. However, we do not believe that our concluding statement in lines 454-455 is different from our explanation here. In lines 454-455, we tried to emphasis the importance of considering 15N signature of input N when interpreting delta 15N of ecosystems as a proxy for ecosystem N cycling.

L381 "It was interpreted" I suppose this was done by the references mentioned just before this sentence. To make this clearer to the reader I suggest to change the sentence from passive into active voice.

Response: We have changed the sentence into active voice.

L393 "in line with our second hypothesis" This can only be true for the first part of this hypothesis

Response: We have separated the hypothesis into two, and wrote it as: ii) N addition would change plant and soil delta 15N towards the 15N signature of the added N due to its incorporation into ecosystem pools, and iii) response of delta 15N to N addition would differ between the two forests due to their difference in their initial N status and N cycling rates. The statement is now related to the second hypothesis.

L396-397 "it shows again" I disagree. From these results one could argue as well that it is the result of increased N availability resulting in increased fractionation plus loss of depleted N.

Response: We understand why the reviewer disagree with the points we made. For plants, fractionation can partly explain the increase in delta 15N, and we have included it in our explanation of the increase in plant delta 15N after the decadal N addition. However, increased fractionation plus loss of depleted N may not explain the decrease in soil delta 15N (although it was not significant) caused by the N addition. The plausible explanation for the changes in delta 15N in both plants and soils is the effect of an imprint from the 15N signature of the added N.

L402 "also after addition of 15N depleted N" In the experiment by the authors the N added was 15N enriched (at least compared to the ambient N deposition).

Response: We referred the added N as '15N-depleted' relative to delta 15N of the bulk soil. This is clarified now.

L440 "calculations based on an isotope mixing model" I would suggest to add some information on how this was calculated and which simplifying assumptions were made in the calculation.

Response: Our wording is not precise enough here, we did a mass balance calculation that uses and assume the added fertilizer as a tracer since its 15N signature differs

from the ambient as well as from both plants and soil. To be meaningful it also requires that significant differences in signatures are observed between the control and the N addition treatment, thus the calculation can only be done for the plant pool. In a previous version we had a long section on this that we now see we had cut the text to be too brief. We have included the assumptions and added more explanations in the text. Including also that for the BF forest the result match that of a tracer experiment that we have published in a separate paper.

L445 "in humid tropical forests of southern China" why would this be true for all these forests, not just for the forest investigated? Possibly because of the delta 15N value of the deposition there (see line 323)? Then the authors should refer to this. Would the region differ in this respect from other regions in the world?

Response: We have already mentioned that forests (including those at our study site and others reported in the references we cited) receive 15N-depleted deposition and thus may also have low delta 15N in plants (written as 'an imprint from 15N-depleted N deposition' in previous version). We have added some words to make it clearer.

L447 "further confirmed" see my remark on L396

Response: We have explained why we focused on the importance of 15N signature of the input N. See our response to the above comments on line 396.

L452 "more important" this is only the case if the 15N signature of the N deposition differs sufficiently from the delta 15N of the ecosystem, and the N deposition is sufficiently large. If that is not the case the mixing probably would not dominate the fractionation plus loss of depleted N.

Response: As shown in our data (Table 2, Table 4), 15N signature of the N deposition differs sufficiently from the delta 15N of the ecosystem. Total N deposition at our site was measured to be 51 kg N per year during 2013-2014, which is sufficiently large. Thus, our conclusion is reasonably sound. We have re-phrased our conclusion to

avoid wording that indicate fractionation is not important at all. We are not sure if the reviewer hints that generalization should be including the cause that he mentioned. We have included that the importance increase with significantly elevated N deposition which is widespread in many regions.

---

## Author Response (AR1)

April 07, 2016

Dear Yakov Kuzyakov,

We are very excited to have been given the opportunity to revise our manuscript, Ms. No. bg-2016-439 "**Nitrogen input $^{15}$N-signatures are reflected in plant $^{15}$N natural abundances of sub-tropical forests in China**". We carefully considered the comments and recommendations offered by the two
referees. Herein, we explain how we revised the paper based on those comments and recommendations.
We want to extend our appreciation for their insightful guidance.

The revision, based on the referees' collective input, includes a number of positive changes. We also got a few suggestions (including language edits) from Professor Knute Nadelhoffer; a native English speaker and a prominent researcher in the field. Based on these guidance, we:

- Edited the abstract to highlight the most important finding of the study in a more flawless language
- Revised portions of the introduction to clearly highlight knowledge gaps, and to clarify the hypotheses to be tested.
- Revised presentation of data, and added more description of the results, as requested
- We have completely revised most of the discussion to accommodate the suggestions and clarify our discussion points

We hope that these revisions improve the paper such that you and the reviewers now deem it worthy of
publication in Biogeoscience. Next, we offer detailed point-by-point responses to comments from the
referees. All line references are referring to the 'marked up version' showing major changes made to the manuscript.

Jiangming Mo (on behave of all co-authors)
Corresponding author

**Response to Anonymous Referee #2**

*The study investigated effects of natural 15N abundance of sources in forest ecosystems on d15N value in two different types of forest ecosystems receiving relatively high nitrogen deposition in China. The study is valuable because there are very few long term nitrogen addition experiments in the area. The theme of the study is suitable for Biogeosciences. However there are some problems and manuscript should be revised.*

**Response**: Thanks for the constructive comments and suggestions that were very useful to improve our manuscript. We have revised our manuscript by implementing those suggested changes and/or by adding more explanation to clarify our points.

**Major comments**

*There are two processes explaining nitrogen isotope ratio; 15N of sources and fractionation processes. Authors discuss the relative contribution of these factors. Authors concluded that source 15N is more important than fractionation in the study. However, it is very difficult to separate these two processes. Authors stressed the importance of source 15N too much. Description of the manuscript should be revised substantially.*

**Response**: We agree that it is difficult to separate the contribution of the two processes. To address this concern, we have revisited our text and conclusions to moderate and clarify the statements on the effect of $^{15}$N of sources, not to oversell the point. Probably our use of the word 'override' is part of overstating the case. In the revision we use 'dominates effects of fractionation' and keep mentioning also the fractionation signal.

*To evaluate the effects of nitrogen addition, nitrogen concentration and d15N values are compered between the control and nitrogen added plots. There only three replication in each treatment and statistical power is very low. Because of this weakness care should be taken when the authors discuss the non-significant results. For example, nitrogen concentration of tree leaves at the pine forest was greater in the N added plot at 10 percent level in table 3. When considering the small number of replication, it is difficult to conclude that there is no significant effect of nitrogen addition.*

**Response**: Lack of enough replication is the common limitation of N addition as well as isotope studies. Not many studies have used three true replicated plots for this kind of studies in tropical forests. We agree that the statistics analysis is not very strong to make strong conclusion. Note however, that for the broadleaf forest 5 dominant tree species were sampled and, since the species differ in %N and $\delta^{15}$N, tree species was included as a random factor in the tests; i.e. for plant compartments the N addition effects build on more than just three determinations. We have carefully checked our wording in the results section to avoid such weakly supported statements.

*In figure 2, delta 15 N in soil seems different between the control and nitrogen added plots, p-value should be shown for each soil layer and total soil as shown for plant compartments. Authors should describe the limitation of the study about statistical analysis and careful interpretations are required.*

**Response**: We have added the data in Fig 2 into Table 4, and the *p*-values asked for are added and shown in the same way as for plants.

**Specific comments**

*L 31, leafs ->leaves*
**Response**: Done

*L 29-30, d15N value of added nitrogen should be described in the sentence.*
**Response**: We have added that the $\delta^{15}N$ of the added N is close to that of atmospheric N (line 23).

*L37, "plant N% was unchanged. . .," nitrogen concentration was marginally increased in pine leaves and significantly in understory vegetation in the pine forest.*
**Response**: We have revised the sentence, indicating the directions (tendency) of changes though it is not significant (lines 28-29). We agree that the term 'unchanged' may not be appropriate for the said reason.

*L39, "the signal from the input may override," 'override' is not a proper word in the situation. Fractionation is also an important process for explaining the difference between plant and soil and between soil depths.*
**Response**: In the revision we used 'dominates the signal' instead of 'override' and keep mentioning also the fractionation effects (see our response above).

*L137, duration of nitrogen addition should be clearly shown.*
**Response**: We have already mentioned that the N addition experiment is ongoing, and it was established in July 2003. To make it clearer, we have mentioned that the N is added monthly since July 2003 (line 136).

*L192-193, information of surface runoff is not sufficient. What is the size of the barrier? How did you collect the water samples?*
**Response**: We have provided more information on how we sampled the surface runoff using the steep slope nature of the plots (lines 176-179).

*L229-230, p-value of statistical analysis should be shown in table 2.*
**Response**: No significant difference was detected between the two forests. This information is now included in the table caption (lines 219-220).

*L235, 'see page' should be 'soil solution.' 15N value of total inorganic N (NH4 plus NO3) should be helpful.*

**Response**: We $\delta^{15}N$ of total inorganic N (NH4 plus NO3)' is now provided. *'see page' changed to 'soil solution'*
*throughout the ms.*

*L238, section title should be revised. It would be "effects of forest types."*
**Response**: Revised as suggested.

*L239-240, the information about earlier study should be described in discussion.*
**Response** : Mention of earlier study moved to the discussion.

*L240-241, clearly indicate that this comparison is about the control plots.*
**Response**: We have indicated in Table and figure captions (line 260, 279 and 296) that comparison is about the control plots. We have also mentioned across the discussion when discussing difference in %N and $\delta^{15}N$ between the two forests (e.g., 407).

*L245-247, p-value should be shown. Information about fig 2 should be included in table 4 as shown for table 3.*
**Response**: The data in Fig.2 is now included into Table 4 as suggested.

*L263, p-value of statistical analysis should be described.*
**Response**: The *p*-values were already given (283-288), hence we deleted this part.

*L295-299, information of the graphs should be included in table 4 and statistical analysis for N addition should be shown.*
**Response**: Changes implemented as suggested.

*L312-316 Figure 3, Information of the graphs should be presented in tables as shown in table 3 and 4. The effects of nitrogen addition should be indicated. Nitrogen concentration and d15N of whole ecosystem (plant plus soil) would be helpful.*
Response: We did not give the combined plant+soil values because i) the soil pool is large and will dominate; ii) the direction of change in $\delta^{15}N$ due to N addition go in opposite direction because the N addition of almost zero $\delta^{15}N$ (of the added N) is right between the level in plants and soil. We decide to keep the figure, since it is better in showing simultaneously the changes in both %N and $\delta^{15}N$.

*L329-332, Mean value of d15N of soil solution is much lower than throughfall or precipitation. Is there any reason for this difference?*
**Response**: Thanks for raising this issue. We have added a discussion of the fractionation pathways including i) nitrification of deposition $NH_4$, that should deplete $NO_3$ product further to be highly negative; ii) nitrification of soil N, where $NO_3$ would still be depleted, but coming from $\delta^{15}N$ 2-5‰, would dilute the negative signal from i); and denitrification of $NO_3$ to

N$_2$ that would enrich the remaining NO$_3$. Since high rates of denitrification have be measured at the site 30 kg/ha/yr (Fang et al., 2015) this may explain why NO$_3$ in soil solution is only slightly depleted. This and other explanation on the observed trend in δ$^{15}$N in input and output fluxes is now discussed in detail (lines 300-325).

*L345-347, when you compare the d15N value between BF and PF, pine forest had lower d15N. The positive correlation between N availability and leaf 15N still exists within the area. Therefore, it is difficult to conclude the results reject the hypothesis.*

**Response**: The reviewer has raised good point here. We want to emphasis that the overall ecosystem $^{15}$N-enrichment in tropical ecosystems due to increased N cycling rate is absent at our study site. Here we compared the $^{15}$N at DHSBR to other tropical forests reviewed by several authors (e.g., Martinelli et al., 1999; Craine et al., 2009; Craine at al., 2015). We have and also re-written the discussion (lines 328-367) and clarified our wording here, including avoiding the use of the word 'overridden' as mentioned above. However, we acknowledge the difference between the two forests that support the hypothesis because the N-rich BF forests are more $^{15}$N-enriched than the somewhat more N-limited PF.

*L362-375, description is only about BF. Is there any comment on PF?*

**Response:** We did not focus on the PF because we observed no clear pattern in soil δ$^{15}$N with soil depth. We believed this to be due to the effects of erosion and soil mixing caused by human disturbances until recent years in the PF. Now we have explained this in section 4.4.

*L382, the contribution of fractionation process and source 15N value is not clearly known in this study.*

**Response**: We agree that the contribution of fractionation process and $^{15}$N signature of source cannot be separated using the current data. The statement was not meant to rule out fractionation but to show that the $^{15}$N signature of source N can explain the change. Again we changed the wording (see our response to the above general comment).

*L392-397, it is difficult to understand. N addition possibly decreases the fractionation during n mineralization and may increase plant 15N. It is difficult to conclude that 15N source is main sole factor.*

**Response**: The globally established knowledge is that increased N input increases fractionation, and we do not have evidence that N addition might have reduced fractionation during mineralization. However, we agree that fractionation can still be important contributing factor and we had no intention to boldly say that '$^{15}$N source is main sole factor'. We have revised the discussion to elaborate about the effects of N addition on δ$^{15}$N of both plants and soils, and how this can be interpreted in terms of the importance of both fractionation and $^{15}$N signature of sources to explain δ$^{15}$N of the studied forests (line 376-378).

*L398-403, the results are based on non-significant results. It is very difficult to conclude the decrease is due to 15N of added N. Because N input by thoughfall has lower value than added N, 15N of total N input should be lower in the control plots. I thought the description is not correct.*

**Response**: We agree with the referee that $^{15}$N of total N input should be lower in the control plots. We assume that $\delta^{15}$N of thoughfall equally affects plants in control and N-plots. Any difference between in $\delta^{15}$N is likely due to the effect of the N addition. Obviously the effect of N addition was not significant due the large soil N pool that is less responsive to contemporary N input manipulation. But, the observed trend draws in the direction of $^{15}$N signature source input.

*L405-442, the section should be moved to just before the previous section 4.2.*
 **Response**: This is solved by rewriting of major portions of the Discussion.

*L415-416, it is difficult to conclude that source 15N is more important in PF. It is too speculative.*
**Response**: Change in plant N content and response of plant $\delta^{15}$N was higher in the PF than in BF, indicating that input N is more important for plant N source in the somewhat more N-limited PF. However, we have revised the part of the discussion to avoid too speculative statements (lines 400-406). Our corrected mass balance calculation showed that similar amount of (~15%) of added N is incorporated into plants pools in both forests.

*L436-439, description about N addition should not be described in this section.*
**Response:** We were not sure what the reviewer meant here. But this part of the text was changed also in response to the other reviewer.

*L445-455, it is difficult to conclude that 15N of source is more important than fractionation process. The contribution of fractionation is still also important factor. Conclusion should be revised substantially.*
**Response:** We have revised the conclusion with careful wording as also detailed in our response to the general comments above.

**Response to W. W. Wessel (Referee)**

General comments
*This paper discusses two processes that affect the delta 15N of forests. Firstly the mixing with N deposition with a different 15N signature than the forest itself and secondly the fractionation of the 15N through different N transformation processes followed by the loss of the lighter fraction resulting in enrichment of the remaining N with 15N. The latter process is thought to happen more strongly if N availability is larger and so it is thought that a higher delta 15N is an indication of a higher N availability. The authors present two sets of delta 15N results of two forests in southern China:*
*the results of the ambient situation ('control') and the results of a long term N addition experiment in the same forests. In the control experiment they find rather low delta 15N values compared to literature values. As the delta 15N value of the substantial N deposition is also rather low, they conclude that the mixing with N deposition with a low delta 15N is the dominating process in determining the 15N of the vegetation and fractionation combined with loss is not important. Secondly they discuss the effects of a long term addition of N with a higher delta 15N than that of the ambient deposition.*

*Here they conclude that the increase in delta 15N in the vegetation is not the result of increased fractionation and loss, due to the higher N availability but that this is the result of the mixing of the added N with a high delta. Their general conclusion is that when delta 15N of forests is used to say something about N availability more attention should be given to the possible influence the delta 15N signature of the N deposition can have.*

*Although I do not think that the main conclusions of the authors are incorrect, I think their argument does need substantial improvement.*

**Response:** Thanks for the constructive comments and suggestions. We have seriously addressed all the concerns. For each general and specific comment, and have provided our responses indicating all the changes we made to the manuscript.

*1. In the first place they do not make clear what delta 15N value they do expect for their forest (under ambient conditions) as a result of fractionation and loss, as described in their hypothesis i (line 101). This hypothesis i is unclear (see below) and they do not explicitly compare this hypothesis with their results. In the Discussion section they compare their results not with individual forests from the literature but with large datasets synthesized from many different forests. Why would their conclusions about their own forests not be true for the forests they cite from the literature? If not, what could be the relevant differences between their forests and those from the literature? Maybe the values calculated for southern China by* **Amundson et al (2003)**, *based on MAP and MAT can help to structure this part of the discussion?*

**Response:** Thank you for the suggestion for using the Amundsen study to initiate the discussion part. We have strived to clarify our hypotheses and their discussion along the lines suggested. Now we have clearly indicated that we expect $\delta^{15}$N value under ambient conditions to be higher for our study forest in our re-phrased hypothesis (line 102-104). We have also compared the hypothesis (all hypotheses) with our results (lines 340-342). The discussion is improved by comparing our results to both individual forests from the literature (temperate forest which we missed in previous version) and large datasets synthesized from many different forests across the world (lines 328-339).

*2. At first sight it seems reasonable to consider mixing to be important in the control experiment, but this could be supported with some calculations of the effect of mixing. It seems the authors have carried out such calculations at least for some cases according to their statement in line 440, but this would be useful for this case as well.*

Response: We did mixing calculations in the control plot using two-source mixing model (Dawson et al. 2002, cited in the previous manuscript), and assuming soil N and deposition N as the major N source to plants. The result showed fraction of N contributed by deposition is 60-80% in the two forests with the higher value being for the pine forest. If plants uptake such large proportion of the deposited N, it likely influence $\delta^{15}$N of plants as we explained in the discussion because deposition is high at our site and it is strongly $^{15}$N-depleted. However, the 60-80% fractions are very high and also ignore the effects of fractionation, so we did not include them.

*3. The reasoning in lines 339-356 is very difficult to follow. I will make more specific comments below.*

**Response:** We have made substantial changes to the texts by comparing our results only to the global data in Martinelli et al. 1999, but also to other studies from temperate forests including those from temperate sites that have high N status (line 339). We also made it clear how our original first hypothesis was rejected by our results (lines 340-341). See our responses to the specific comments too.

*4. Concerning the N addition experiment it can be said that both the mixing process and the increased fractionation plus loss process (expected as a result of larger N availability) would lead to an increase in the delta 15N of the vegetation, so it is unclear why the authors choose that the increased delta 15N values found in the vegetation were the result of mixing and not of increased fractionation plus loss. What result of the experiment and the measurements would have led them to the other conclusion?*

*In fact probably both mixing and fractionation plus loss contribute to some extent to the increase of delta 15N in the vegetation. Again some calculations of the mixing of the deposition might give more insight into the potential contribution of this process.*

**Response:** We agree that fractionation combined with loss of $^{15}$N-depleted N can be caused by larger N availability, and it can be one of those important factors that control ecosystem $\delta^{15}$N. The increase in plant $\delta^{15}$N after the N addition can be partly explained by this fractionation plus loss processes. However, the same fractionation may not explain the tendency of decrease in soil $\delta^{15}$N. When we used the added N with -0.7 delta label as tracer for mass balance calculation (Nadelhoffer and Fry, 1994) about 15% of the added N was estimated to be taken up by the plants in both forests, and thus hint that the input N is substantially incorporated into plants although they over all do not increase the uptake in BF. See our response to similar question from the other referee.

*5. I think there is something wrong with the statistical results presented in Tables 3 and 4. The tests for siginifant differences sometimes yield significant p values while the difference tested is smaller than the sum of the two standard errors. This cannot be correct. I suggest the authors provide the data and the script they have used to calculate the statistics so it becomes clear what they have done. See for example in Table 3 twigs difference between BF and PF is 0.29, while the sum of the SEs is 0.96*

*and p<0.01 and in Table 4 tree leaf in BF difference between control and N addition is 0.6, while the sum of the SEs is 1.1 and p<0.01. I assume two-sided tests were carried out although this was not mentioned.*

**Response:** For the comparison of $\delta^{15}$N between the two forests, we have mentioned that *t*-test was used as the reviewer assumed it, and the section about statistics is edited to show this (lines 195). In Table 3, we had made a copy-past mistake; the correct SE for twigs in PF is 0.05, not 0.77, and we have corrected it. Further, it is important to note (as mentioned in the table heading) that for the broad-leaved forest, 5 dominant tree species were sampled and since the species differ in %N and $\delta^{15}$N, species was included as a random factor in the tests using mixed ANOVA (mentioned in section 2.5); i.e. for plant compartments the N addition effects build on more than just three measurements. Thus, the overlap of the SE's based plot means may not be instructive in that case.

*6. I would suggest that the authors should be more careful in using the terms 15N enriched and 15N depleted and define what exactly is meant by them and relative to what (below or above zero, or relative to the delta of some other pool or flux). They use these terms many times throughout the text. See e.g. my comment below on line 402. In line 32 even the term "more enriched" is used.*

**Response:** We have strived to clarify the wording and have in some cases added the changes in $\delta^{15}N$ to improve on this.

**Specific comments**

*L25 "examined the measurement": this suggests the paper is about measurement techniques. I suggest to rephrase this.*

**Response:** We have rephrased the sentence.

*L31 "leafs" the text contains many spelling errors; I suggest the authors check the text throughout for these.*

**Response**: We have changed 'leafs' to 'leaves'. Similar spelling errors were thoroughly searched for and corrected.

*L31: "old-growth forest" this forest is everywhere else described as broadleaved forest, so I would suggest to use that term here as well*

**Response**: Corrected as suggested.

*L48 "recently" I think it is relevant to be more specific, so the reader knows how long this N addition has been going on. In the methods the 1990s are mentioned for DHSBR (L113).*

**Response**: We have made it more specific, and mentioned that increase in N deposition in China has been increasing continuously since the 1980s (Liu et al. 2011) (lines 59).

*L67 "above the atmospheric standard" I wonder whether for this criterion 0.0 0/00 is the relevant value, as atmospheric N2 is not a direct source of N for a terrestrial ecosystem.*

**Response**: We have deleted the 'above the atmospheric standard' because the atmospheric N may not be direct source of N to the plants.

*L81-82 "hotspots" If this is meant to be high in N deposition, I would suggest to use the latter term.*

**Response**: We have deleted 'hotspots' and used 'high N deposition' (line 81).

*L102 The comparison of a high N forest with temperate forests seems inconsequent. What about temperate forests with high N status?*

**Response:** We have re-phrased our hypothesis to indicate that the comparison includes temperate forests in general (not only N-limited forests) (lines 102-104). Similarly, the discussion is improved by adding some studies from different temperate forests that cover wider gradient of N availability including N-saturated forests (e.g., Koopmans et al., 1997; Sah & Brumme, 2003) (line 339).

*L103 second hypothesis: I wonder which results could lead to the rejection of this hypothesis, given the experimental conditions. The first part of this hypothesis seems not very challenging and the second part is not very specific.*

**Response:** We have separated the hypothesis into two (see our response to your comment on 393). We have referred each of our three hypotheses in the discussion and explained if they are confirmed or rejected by our results.

*L114 "steep slopes" Amundson et al (2003) have suggested that under these circumstances delta 15N might be lower (see their paragraph [26])*

**Response:** We are aware that topography can have a significant influence on the landscape-scale patterns of plant and soil $\delta^{15}N$. The preliminary data set presented in Amundson et al (2003) was from the central California coast range (Figure 4), suggested that topography could explain up to 2% variation in soil $\delta^{15}N$ at a given location. Apart from these observation used to explain within site variation in soil $\delta^{15}N$, we found no study that compared sites with different topography. We do not believe this steep slope at DHSBR is important factor to explain the distinct $^{15}N$-depletion compared to other tropical forests. But we have mentioned this as a potential (minor) influence in the discussion (lines 364-367).

*L182 "including a dry period" If the authors mean that there were not any water samples in Dec and Jan because of a lack of precipitation then please state this.*

**Response:** We have indicated that these dry months are the period when there were not any water samples because of a lack of precipitation (lines 169).

*L186-187 A collector with an area of 8000cm2 seems extremely large. Is this a correct value?*

**Response**: Corrected to show that $0.8m^2$ was a total interception area for the five collectors (173).

*L215 "plant species as a random factor" Apparently this is not the case for the pine forest, which contains only one dominant species (L159).*

**Response**: Mixed model ANOVA was where plant species used as a random factor was used for compartments with mean from several species (canopy tree in BF and understory vegetation in both forests. For canopy layer in PF, only one dominant species, a simple t-test was used. We have clarified this in the manuscript (lines 195-198).

*L233 Table 2 . At first sight it looks like leaching losses have lower deltas than the deposition, indicating the occurrence of fractionation and loss of N with low deltas, thus increasing the 15N content of the remaining N. However, as deposition is dominated by NH4, while leaching is dominated by NO3, this is not the case. Calculating a weighted average delta 15N for all chemical species in all fluxes may show this. This can support the argument that fractionation plus loss is not evident from this budget, although it is of course incomplete. Are here not any values for the added N plots?*

**Response**: Using $\delta^{15}N$ data in Table 2 and the concentration data presented in Fig. S1 (now Table S1), we calculated the weighted average $\delta^{15}N$ for total dissolved inorganic N (DIN). The data (presented in revised Table 3) showed that soil solution ass slightly higher $\delta^{15}N$ than the deposition in both forests. We have two samples for the added N plots but only in the broad-leaved forest (BF), and it shows even higher $\delta^{15}N$ (~ -2.8‰) than the values in the control plot (-5.7‰, Table 2). Since nitrate is enriched in soil solution, it may appear to show no evidence for fractionation and loss of N with low $\delta^{15}N$. However, this argument may not be true if denitrification (reported at our site) dominates nitrification because the two processes have antagonistic effect on $\delta^{15}N$ of the nitrate. The nitrate measured in the soil solution also comes from nitrification of soil ammonium, which can still be enriched compared to the nitrate in deposition N. These points are now added to the discussion about $\delta^{15}N$ of the water samples (lines 316-325). Also see our response to comments on L329-332 by the other referee.

*L233 Table 2 In the text it is stated that runoff was measured only in one plot per treatment (line 192), so how can there be an average of three measurements for runoff here?*

**Response**: The runoff was collected in one plot per treatment, but at three different points, which we used as replicate (line 179). We have indicated in Table 2 footnote that the SE for runoff SE is for pseudo-replicates within one plot (line 222).

*L251 Fig.1 Are these samples that were taken monthly between Sept and Feb (4 samples) with 3 replicates, in total 12? I suggest to explain this in the caption. Again how was this for runoff (see my previous remark on Table 2)? What could be the cause for the variation found? Is there no substantial time delay between the moment of deposition and the moment the deposited N reaches the subsoil or the runoff?*

**Response**: Yes, these are samples that were taken monthly between Sept and Feb (4 samples) with 3 replicates, making total of 12 samples. For the surface runoff, sampling was done only in one plots, but at three points. However, we decided to only mean from the pseudo-replicates for the four months. We have explained this in the caption line 238-239).

*L261-263 "N concentration of N pool weighted average plant pools calculated per plot". The reader is referred to Table 3, but in there are only N concentrations of individual pools.*

**Response**: This sentence is now deleted because it was misplaced. Data on effects of N addition on N pool weighted average plant pools was correctly presented later and appropriately referred to Fig 2 (which was Fig 3 in the previous version) (lines 283-288).

*L295 Fig.2 I suggest to increase the size of the symbols in the legend so the different patterns used are more easily recognized. This is also a problem in the supplement figure.*

**Response**: Information in Fig 2 is now included in Table 4 based on a comment from another reviewer. We have increased the size of the symbols in all other figurers including the supplement figure.

*L307 "decrease as expected" It is true that the delta of the N input into the forest is still lower than the delta of the soil, but the addition has substantially increased the delta of the total N input, so one might as well expect an increase in the soil delta as a result of this.*

**Response**: Since look at the effect of the N addition we the decrease (from control to N-plot) can only be an effect of the addition, so the $\delta^{15}$N of the total N input should not be relevant here.

*L325 "other regions" please specify which regions are meant.*

**Response**: Other regions mean those in Germany (Freyer 1978) and Chesapeake Bay (Russel et al., 1998). However, we have no evidence for $^{15}$N-depleted deposition N in these regions, and the two cited papers are also old studies. So we have deleted this part of the sentence and explained our finding in relation to other relevant studies.

*L337 "surprisingly" I suggest the authors clarify what they expect here.*

**Response**: Our expectation, as stated in our first hypothesis, was an enrichment of leaf $\delta^{15}$N at our study site that was higher than the average leaf $\delta^{15}$N observed for temperate forests on global scale.

*L342-345 This remark on the enrichment factor seems misplaced here, as nowhere else in the paper something is said about the enrichment factor. It is also unclear to me why this would support the previously mentioned hypothesis.*

**Response**: We agree with this point, and have deleted the sentence.

*L345 "rejects this hypothesis" Which result precisely makes the authors decide to reject? Do the authors reject the full hypothesis or only mean that the increase in delta 15N simply does not happen? Nothing is said about hypothesis i from the introduction. I would suggest to refer to this hypothesis as well, although it needs to be rephrased, as I mentioned earlier.*

**Response**: The hypothesis was that tropical forests that have high soil N availability due to increased N deposition have higher $\delta^{15}$N compared pre-dominantly N-limited temperate forests due to increased fractionation combined with loss of $^{15}$N-depleted N in tropical forests. Since we did not observe such ecosystem enrichment at our site, which also has high N deposition, we concluded that the hypothesis is rejected. To make it clearer, we have re-phrased the whole sentence, and indicated why our result rejects this hypothesis (lines 340-341). We have also re-structured the paragraphs, and re-phrased the hypothesis to make it brief and direct (lines 102-104).

*L347 "other depleting factors" I think "other" should be removed as the previously mentioned process is an enriching factor.*

**Response**: The sentence is re-phrased in connection with our response to the above comment on line 345 (lines 348).

*L348-349 "in other Chinese forests with high N deposition" Why only or especially in Chinese forests? And would this not depend on the delta 15N value of the N deposition? Maybe the authors have the literature in mind they mention in lines 323Â . If that is the case they should refer explicitly to these results. The authors make a different and more general statement in lines 454-455.*

**Response**: Yes, we refer to those for which we already cited references (Fang et al., 2011a; Wang et al., 2014), and we have edited the sentence to show this (lines 345-347). However, we do not believe that our concluding statement in lines 454-455

in previous version (now in lines 435-436) is different from our explanation here. Our conclusion emphasizes the importance of considering $^{15}$N signature of input N when interpreting $\delta^{15}$N of ecosystems as a proxy for ecosystem N cycling, and we have edited the sentence to make it clear (lines 435-436).

*L381 "It was interpreted" I suppose this was done by the references mentioned just before this sentence. To make this clearer to the reader I suggest to change the sentence from passive into active voice.*
**Response**: We have changed the sentence into active voice (lines 390-391).

*L393 "in line with our second hypothesis" This can only be true for the first part of this hypothesis*
**Response**: We have separated the hypothesis into two, and wrote it as:
ii) N addition would change plant and soil $\delta^{15}$N towards the $^{15}$N signature of the added N due to its incorporation into ecosystem pools, and
iii) response of $\delta^{15}$N to N addition would differ between the two forests due to differences in their initial N status and N cycling rates. The statement is now related to the hypothesis number (ii) (lines 394-396).

*L396-397 "it shows again" I disagree. From these results one could argue as well that it is the result of increased N availability resulting in increased fractionation plus loss of depleted N.*
**Response**: We understand why the reviewer disagree with the points we made. For plants, fractionation can partly explain the increase in $\delta^{15}$N, and we have included it in our explanation of the increase in plant $\delta^{15}$N after the decadal N addition (376-378). However, increased fractionation plus loss of depleted N may not explain the decrease in soil $\delta^{15}$N (although it was not significant) caused by the N addition. The plausible explanation for the changes in $\delta^{15}$N in both plants and soils is the effect of an imprint from the $^{15}$N signature of the added N. We have revised the statement with clearer explanation indicating how the importance of $^{15}$N signature of sources outweighs that of fractionation in soil (lines 389-393).

*L402 "also after addition of 15N depleted N" In the experiment by the authors the N added was 15N enriched (at least compared to the ambient N deposition).*
**Response**: We referred the added N as '$^{15}$N-depleted' relative to $\delta^{15}$N of the bulk soil. We have clarified this as a relative comparison, and presented the $\delta^{15}$N for the added N and plants (line 378).

*L440 "calculations based on an isotope mixing model" I would suggest to add some information on how this was calculated and which simplifying assumptions were made in the calculation.*
**Response**: Our wording is not precise enough here, we did a mass balance calculation that uses and assume the added N as a tracer since its $^{15}$N signature differs from the ambient deposition N as well as that of both plants and soil. For the calculation to be meaningful, it requires that $\delta^{15}$N significant differ between the control and the N-plots, thus the calculation can only be done for the plant pool. We have cut the long text on this in the previous version to make it brief and easy to understand, have included the assumptions that were made in the calculation (line 379-388).

*L445 "in humid tropical forests of southern China" why would this be true for all these forests, not just for the forest investigated? Possibly because of the delta 15N value of the deposition there (see line 323)? Then the authors should refer to this. Would the region differ in this respect from other regions in the world?*

**Response**: We have already mentioned that forests (including those at our study site and others reported in the references we cited) receive $^{15}$N-depleted deposition and thus may also have low $\delta^{15}$N in plants (written as 'an imprint from $^{15}$N-depleted N deposition' in previous version). We have added some words to make it clearer (line 429).

*L447 "further confirmed" see my remark on L396*

**Response**: We have explained why we focused on the importance of $^{15}$N signature of the input N. See our response to the above comments on line 396. We have made slight changes in our wording (e.g., used 'support' instead of 'confirmed') not to overstate the conclusion (lines 429-432).

*L452 "more important" this is only the case if the 15N signature of the N deposition differs sufficiently from the delta 15N of the ecosystem, and the N deposition is sufficiently large. If that is not the case the mixing probably would not dominate the fractionation plus loss of depleted N.*

**Response**: As shown in our data (Table 2, Table 4), $^{15}$N signature of the N deposition differs sufficiently from the $\delta^{15}$N of the ecosystem. Total N deposition at our site was measured to be 51 kg N per year during 2013-2014, which is sufficiently large. Thus, our conclusion is reasonably sound. We have re-phrased our conclusion to avoid wording that indicate fractionation is not important at all (line 434). We are not sure if the reviewer hints that generalization should be including the cause that he mentioned. We have also included that the importance increase with significantly elevated N deposition which is widespread in many regions (lines 435-436).

[revised manuscript text omitted]